# CORE: COMMON RANDOM RECONSTRUCTION FOR DISTRIBUTED OPTIMIZATION WITH PROVABLE LOW COMMUNICATION COMPLEXITY

## ABSTRACT

With distributed machine learning being a prominent technique for large-scale machine learning tasks, communication complexity has become a major bottleneck for speeding up training and scaling up machine numbers. In this paper, we propose a new technique named Common randOm REconstruction (CORE), which can be used to compress the information transmitted between machines in order to reduce communication complexity without other strict conditions. Especially, our technique CORE projects the vector-valued information to a low-dimensional one through common random vectors and reconstructs the information with the same random noises after communication. We apply CORE to two distributed tasks, respectively convex optimization on linear models and generic non-convex optimization, and design new distributed algorithms, which achieve provably lower communication complexities. For example, we show for linear models CORE-based algorithm can encode the gradient vector to $\mathcal{O}(1)$-bits (against $\mathcal{O}(d)$), with the convergence rate not worse, preceding the existing results.

## 1 INTRODUCTION

Distributed machine learning and optimization have become the main technique for solving tasks with large model and data scales. In simple terms, the distributed optimization problem in machine learning can be regarded as minimizing an objective function $f$ defined as an average of individual functions that are respectively accessible by their corresponding local machines. More specifically, we consider a constrained optimization problem

$$\underset{\mathbf{x}\in\mathbb{R}^d}{\text{minimize}} \quad f(\mathbf{x}) \equiv \frac{1}{n}\sum_{i=1}^n f_i(\mathbf{x}_i) \tag{1}$$
$$\text{s.t. } \mathbf{x}_1 = \mathbf{x}_2 = \cdots = \mathbf{x}_n.$$

Here $f_i$ represents the individual objective function at the local machine $i$ and the constraint in (1) guarantees different machines corporately finding the same minimizer of the global objective function $f$. Typical examples for $f_i$ include regression or classification over linear, graphic, as well as (deep) neural network models. In these cases, $f_i$ shares the form as $f_i(\mathbf{x}) \equiv F(\mathbf{x};\zeta_i)$, where $\zeta_i$ denotes the data stored in machine $i$ and $F$ represents the learning model.

One dominating **bottleneck** for further improving the speed of distributed machine learning is the communication bandwidth. With the increase of machine numbers and parameter scale, time spent on communication can not be ignored and even becomes much longer than that on computation. Such a problem is much more salient when the bandwidth of computing cluster is restricted, such as mobile devices. Many researchers have noticed that reducing the dimensions of data transmitted between machines can effectively reduce the communication complexity, and proposed heuristic techniques, such as quantization (Seide et al., 2014) and sparsity (Aji & Heafield, 2017), to reduce the communication burden to some degree. Some more complete and theoretically guaranteed algorithms based on these techniques are proposed soon, However, to the best of our knowledge, although some researches show how to improve existing compression techniques or propose several new ones, few results provide concrete and feasible compression techniques that can provably reduce communication costs and maintain algorithm accuracy under mild conditions. In this paper,

we propose a new technique named Common randOm REconstruction (CORE) which presents a provable result on low communication complexity. CORE is a technique that can be used to transmit a sequence of vector-valued information that follows from well-known ideas from information theory and communication complexity theory, taking advantage of common random variables. At each round, the vector-valued information is projected to a low-dimensional vector using Gaussian random noises by the sender, and after communication reconstructed with the same noises by the receiver. We show such a procedure generates an unbiased estimator of the original vector-valued information with a controlled variance. We apply CORE to two distributed tasks, namely convex optimization on linear models and generic non-convex optimization. Compared with some existing relevant researches, ours has certain advantages. First, we propose a concrete and feasible compression technique and algorithms instead of an abstract but potentially not implementable framework to reduce communication costs. Second, our algorithms provably achieve much lower communication costs compared with the existing algorithms under realizable conditions.

## 1.1 RELATED WORK

In this section we briefly introduce the related work about our methods, including gradient compression, random sketching technique, distributed optimization and federated learning, and random communication complexity. A more detailed introduction can be seen in Appendix A.

**Gradient compression.** Gradient compression is the main technique to reduce communication complexity during the process of training. The representative achievements are gradient quantization (Seide et al., 2014; Tang et al., 2021) and gradient sparsification (Wangni et al., 2018; Shi et al., 2019; Jiang & Agrawal, 2018). Moreover, some methods (Wen et al., 2017; Alistarh et al., 2017; Wu et al., 2018; Faghri et al., 2020; Horvóth et al., 2022; Mishchenko et al., 2019; Aji & Heafield, 2017; Lin et al., 2017; Wang et al., 2018; Mishchenko et al., 2020) obtained better results based on previous works. In addition, some new techniques based on innovative ideas have also been developed and achieved good results. For example, PowerSGD (Vogels et al., 2019) proposed a new low-rank gradient compressor. Other techniques (Bernstein et al., 2018; Safaryan & Richtárik, 2019; Beznosikov et al., 2020; Horváth et al., 2023; Richtárik et al., 2022) were also proposed as innovative new achievements. However, the second moments of these estimations are often of order $d$, which implies a restriction of the total communication costs.

**Random sketching.** Sketching (Gribonval et al., 2020; Woodruff et al., 2014; Ikonomovska et al., 2007) is a widely-used technique in machine learning, data mining and optimization, whose core idea is to reduce the scale by a probabilistic data structure to approximate the data to reduce the computation costs. It is worth noticing that some researchers have started to use the sketching technique to reduce communication costs during the process of training. For example, FedAvg (Konečnỳ et al., 2016) and SKETCHED-SGD (Ivkin et al., 2019), which uses Count Sketch (Charikar et al., 2004) to compress the gradient. They also presented a theoretical analysis of convergence, but when $d$ is large, it is much worse than SGD. Hanzely et al. (2018) proved that when adding biased estimates on the basis of random matrix sketching, their algorithm achieves a faster convergence rate and can be accelerated. However, they did not come up with a specific sketching method. Moreover, Lee et al. (2019) and Pilanci et al. (2015) proposed some sketched Hessian-based second-order optimization algorithms. In this work, we mainly focus on gradient-based communication-efficient methods.

**Distributed optimization.** Distributed machine learning and optimization have developed rapidly in recent years. In the early years, the main achievements were based on the existing optimization algorithms (Cotter et al., 2011; Lee et al., 2015; Shi et al., 2015; Scaman et al., 2017b). In recent years, some compressed gradient descent algorithms (Khirirat et al., 2018; Mishchenko et al., 2019; Gorbunov et al., 2021; Tyurin & Richtárik, 2022; Li & Richtárik, 2021) based on compression techniques mentioned above were also proposed. But almost all the methods above have the total communication costs at $\mathcal{O}(d)$ level. It is worth noticing that in practice $d$ is often extremely large. So there is still a lack of a concrete compression technique and corresponding distributed algorithm that achieves low communication complexity when $d$ is large. Our work fills this gap. In addition, error feedback technique (Stich & Karimireddy, 2019; Karimireddy et al., 2019; Tang et al., 2019; Gruntkowska et al., 2022; Richtárik et al., 2021; Fatkhullin et al., 2021) was also widely used in compressed distributed optimization.

**Federated learning.** Federated Learning is another machine learning setting concentrating on communication costs, where the goal is to train a high-quality centralized model while training data remains distributed over a large number of clients each with unreliable and relatively slow network connections. In the early years, some federated learning algorithms (Konečnỳ et al., 2016; Rothchild et al., 2020; Ivkin et al., 2019; Karimireddy et al., 2020; Mitra et al., 2021) based on the local gradient have been proposed. However, the approximation of local gradient often results in a loss of convergence rate. The total communication costs are either worse than or equal to those of vanilla gradient descent. Recently, some new communication-efficient methods such as Scaffnew (Mishchenko et al., 2022) and GradSkip (Maranjyan et al., 2022) have been proposed to achieve the same communication rounds as the lower bound of smooth and strongly-convex objective functions $\mathcal{O}(\sqrt{\kappa})$, but the total communication costs are still $\mathcal{O}(d)$.

**Random communication complexity.** In theoretical computer science, communication complexity studies the amount of communication needed to solve a problem when input data is distributed among several parties. Communication complexity was first proposed in Andrew (1979). Andrew (1979) also defined randomized protocol and randomized communication complexity. In a randomized protocol, parties are given a common random string as the input to a deterministic protocol. Random protocols can determine the answer in high probability with much less amount of information transmitted, so randomized communication complexity is much lower than deterministic communication complexity in expectation. Inspired by the advantage of randomized protocols over deterministic ones, we designed a random compression method for distributed optimization which is faster in expectation. Newman (1991) proved that any protocol using a common random string can be simulated by a private random string protocol, with an extra $\mathcal{O}(\log n)$ bits.

## 1.2 CONTRIBUTIONS

In this work, we introduce the Common randOm REconstruction (CORE) technique and demonstrate its application in two distributed tasks. The advantages of utilizing CORE in these tasks are outlined below.

To the best of our knowledge, CORE is the first concrete and feasible compression method that achieves a limited bounded variance of the estimate and provably reduce communication complexity when the eigenvalues of the Hessian matrices of $f$ drop very fast. We have observed that in practice, the rapid decrease of eigenvalues in the Hessian matrix has long been recognized. For instance, researchers have introduced concepts like effective rank (e.g., Hsu et al. (2012)) to quantify the dimensionality of the data's influence on linear models. Some recent empirical studies (Sagun et al., 2016) carefully compute the eigenvalue of Hessian curves during training for (deep) neural networks. (See Figure 4 for an example of eigenvalues of a real dataset and a neural network in Appendix L).

To characterize the strength of CORE in rigor, we introduce the factor

$$r_\alpha = \sup_{\mathbf{x} \in \mathbb{R}^d} \sum_{i=1}^{d} \lambda_i^\alpha(\nabla^2 f(\mathbf{x})), \qquad \alpha > 0 \tag{2}$$

as the effective dimension for distributed optimization, where $\lambda_i(\cdot)$ is the $i$-th singular value (also the eigenvalue when $\nabla^2 f(\mathbf{x})$ is semi-definite in convex case). This is inspired by the recent work of zeroth-order optimization (Yue et al., 2023), Langevin sampling (Freund et al., 2022), and distributed optimization (Hanzely et al., 2018). We further introduce the Hessian domination assumption, a concept employed in various studies for theoretical analysis (Hanzely et al., 2018; Safaryan et al., 2021; Yue et al., 2023). We apply CORE to some gradient-descent-based algorithms and use the effective dimension $r_\alpha$ to characterize their communication costs. By combining CORE with centralized gradient descent (CGD), we propose the CORE-Gradient Descent (CORE-GD) algorithm for linear regression and prove that for the standard case where $f$ has $L$-Lipschitz gradients, CORE-GD achieves $\mathcal{O}\left(r_1(f)D^2\epsilon^{-1}\right)$ communication costs to obtain an $\epsilon$-optimal solution, where $D = \|\mathbf{x}^0 - \mathbf{x}^*\|$. Compared with CDG which achieves $\mathcal{O}\left(dLD^2\epsilon^{-1}\right)$ communication costs, CORE-GD has a significant advantage since $r_1(f)$ is much smaller than $dL$ in most cases when eigenvalues decay fast. In Appendix B, we also study accelerations of CORE-GD using the momentum technique, and propose a heavy-ball-based accelerated algorithm named CORE-Accelerated Gradient Descent (CORE-AGD) for linear regression. We prove that CORE-AGD achieves the state-of-the-art $\tilde{\mathcal{O}}\left(\frac{r_{1/2}(f)}{\mu^{1/2}}\right)$

Table 1: The performance of communication-efficient methods

| method | communication rounds | compressor | floats sent per round | total communication costs |
|---|---|---|---|---|
| CGD Nesterov (2003) | $\tilde{\mathcal{O}}(\frac{L}{\mu})$ | - | $\Theta(d)$ | $\tilde{\mathcal{O}}(\frac{dL}{\mu})$ |
| ACGD Nesterov (2003) | $\tilde{\mathcal{O}}(\frac{L^{1/2}}{\mu^{1/2}})$ | - | $\Theta(d)$ | $\tilde{\mathcal{O}}(\frac{dL^{1/2}}{\mu^{1/2}})$ |
| FedLin Mitra et al. (2021) | $\tilde{\mathcal{O}}(\frac{d^{3/2}L}{k^{3/2}\mu})$ | Top-K [1] | $\Theta(k)$ | $\tilde{\mathcal{O}}(\frac{d^{3/2}L}{k^{1/2}\mu})$ |
| Scaffnew Mishchenko et al. (2022) | $\tilde{\mathcal{O}}(\frac{L^{1/2}}{\mu^{1/2}})$ | Skip [2] | $\Theta(d)$ | $\tilde{\mathcal{O}}(\frac{dL^{1/2}}{\mu^{1/2}})$ |
| GandSkip Maranjyan et al. (2022) | $\tilde{\mathcal{O}}(\frac{L^{1/2}}{\mu^{1/2}})$ | Skip [2] | $\Theta(d)$ | $\tilde{\mathcal{O}}(\frac{dL^{1/2}}{\mu^{1/2}})$ |
| DIANA Mishchenko et al. (2019) | $\tilde{\mathcal{O}}(\frac{d}{K} + \frac{dL}{Kn\mu})$ [3] | Top-K [1] | $\Theta(K)$ | $\tilde{\mathcal{O}}(d + \frac{dL}{n\mu})$ |
| ADIANA Li et al. (2020) | $\tilde{\mathcal{O}}(\frac{d}{K} + \frac{dL^{1/2}}{Kn^{1/2}\mu^{1/2}})$ [3] | Top-K [1] | $\Theta(K)$ | $\tilde{\mathcal{O}}(d + \frac{dL^{1/2}}{n^{1/2}\mu^{1/2}})$ [4] |
| ASEGA Hanzely et al. (2018) | $\tilde{\mathcal{O}}(\frac{\sum_{i=1}^{d} A_{ii}^{1/2}}{\mu^{1/2}})$ | - | $\Theta(1)$ [5] | $\tilde{\mathcal{O}}(\frac{\sum_{i=1}^{d} A_{ii}^{1/2}}{\mu^{1/2}})$ |
| CORE-GD (this work) | $\tilde{\mathcal{O}}(\frac{L}{\mu})$ | CORE | $\Theta(\frac{\text{tr}(\mathbf{A})}{L})$ | $\tilde{\mathcal{O}}(\frac{\text{tr}(\mathbf{A})}{\mu})$ |
| CORE-AGD (this work) | $\tilde{\mathcal{O}}(\frac{L^{1/2}}{\mu^{1/2}})$ | CORE | $\Theta(\frac{\sum_{i=1}^{d} \lambda_i^{1/2}}{L^{1/2}})$ | $\tilde{\mathcal{O}}(\frac{\sum_{i=1}^{d} \lambda_i^{1/2}}{\mu^{1/2}})$ |

[1] FedLin, DIANA and ADIANA only propose the algorithms using compressor, but do not propose concrete gradient compression technique. They use Top-K as an example to analyse the communication rounds and costs.

[2] Scaffnew and GandSkip use communication skipping instead of gradient compressor. Specifically, they only communicate every $\mathcal{O}(\frac{L^{1/2}}{\mu^{1/2}})$ rounds and the total computation rounds are $\tilde{\mathcal{O}}(\frac{L}{\mu})$.

[3] The communication rounds of DIANA are $\tilde{\mathcal{O}}(\omega + \frac{\omega L}{n\mu})$ when $\omega \geq n$. And similarly, that of ADIANA is $\tilde{\mathcal{O}}(\omega + \frac{\omega L^{1/2}}{n^{1/2}\mu^{1/2}})$ when $\omega \geq n$. Here $\omega$ is compression ratio. For example, when using Top-K compressor, the compression ratio is $\frac{d}{K}$, which is much larger than $n$ when the dimension of data is extremely large. In this setting $n$ can be seen as $\mathcal{O}(1)$.

[4] The theoretical bound of the total communication costs of this method is $\tilde{\mathcal{O}}(d + \frac{d^{1/2}L^{1/2}}{\mu^{1/2}})$, and the bound of CORE-AGD is $\tilde{\mathcal{O}}(\frac{d^{1/2}\text{tr}(\mathbf{A})^{1/2}}{\mu^{1/2}})$. In most cases when $\text{tr}(\mathbf{A})$ is bounded and $d$ is much large, CORE-AGD is better.

[5] This method is coordinate-descent-based. We show that CORE-AGD is theoretically better. Letting $\mathbf{A} = \mathbf{U}^\top \Sigma \mathbf{U}$ where $\mathbf{U} = [u_{ij}]$ and $\Sigma = \text{diag}\{\lambda_i\}$, we have $A_{ii} = \sum_{j=1}^{d} \lambda_j u_{ji}^2 \geq (\sum_{j=1}^{d} \lambda_j^{1/2} u_{ji}^2)^2$ (because the Hessian matrix is positive definite and symmetric). Thus we have $\sum_{i=1}^{d} A_{ii}^{1/2} \geq \sum_{i=1}^{d} \lambda_i^{1/2}$.

communication costs which is lower than $\tilde{\mathcal{O}}(d + \frac{dL^{1/2}}{n^{1/2}\mu^{1/2}})$ in Li et al. (2020) and $\tilde{\mathcal{O}}\left(\frac{\sum_{i=1}^{d} M_{ii}^{1/2}}{\mu^{1/2}}\right)$ in Hanzely et al. (2018). More details and comparisons are shown in Table 1. Compared with the results in Hanzely et al. (2018), our works present a concrete compression technique. In Section 5, we then examine the efficiency of CORE in generic non-convex optimization when finding an $\epsilon$-approximated first-order stationary point. We further assume a Hessian-Lipschitz condition and show that CORE-GD with carefully chosen stepsize can achieve lower communication costs which reduces upon the communication costs of CGD by a $\min\left\{dL/r_1(f), \epsilon^{-0.5}d^{1/4}\right\}$ factor.

In summary, the contribution of the paper is listed below:

(A) We propose a new technique called CORE to efficiently transmit information between machines. To the best of our knowledge, CORE is the *first* concrete and feasible compression technique that is provably more efficient on communication when eigenvalues drop fast and can be applied to gradient-descent-based algorithms.

(B) We apply CORE to convex optimization on linear models and generic non-convex optimization. We design new optimization algorithms and show a *remarkable reduction* of communication complexity under realizable conditions. Compared with the recent distributed optimization and federated learning algorithms, our CORE-GD and CORE-AGD achieve the lower bound of iteration rounds the *state-of-the-art* total communication costs under the realizable condition.

Finally, we propose a reduction framework that extends CORE to work on decentralized communication in Appendix E. We show the price is only an additional $\tilde{\mathcal{O}}(\sqrt{\gamma})$ factor, where $\gamma$ is the eigengap of the gossip matrix for the network topology. We also show that CORE is equipped with some privacy guarantee naturally for the use of random vectors, and prove our results in Appendix J. We conduct empirical studies where we compare CORE with the basic frequently used quantization and sparsity techniques both on linear models and (deep) neural networks in Appendix K.

## 1.3 NOTATION

Throughout this paper, we use the convention $\mathcal{O}(\cdot)$, $\Omega(\cdot)$, and $\Theta(\cdot)$ to denote the *lower*, *upper* and *lower and upper* bound with a global constant, and use $\tilde{\mathcal{O}}(\cdot)$ to denote the lower bound that hides

a poly-logarithmic factor of the parameters. Let $\mathbb{R}$ denote the set of real numbers, and $\mathbb{R}^d$ denote a $d$-dimensional Euclidean space. We use bold lowercase letters, like $\mathbf{x}$, to represent a vector, and bold capital letters, like $\mathbf{A}$, to represent a matrix. Specially, we use $\mathbf{I}_d$ to represent the identity matrix in $d$-dimensional Euclidean space, and omit the subscript when $d$ is clear from the context for simplicity. Let $\langle \cdot, \cdot \rangle$ denote the inner product of two vectors in the Euclidean space, $\|\mathbf{x}\|$ denote the Euclidean norm of a vector, and $\|\mathbf{A}\|$ denote the operator norm of a matrix. It is worth noticing that we use $\|\mathbf{x}\|_{\mathbf{A}}$ to denote the Mahalanobis (semi) norm where $\mathbf{A}$ is a positive semi-definite matrix, which can be specifically defined as $\|\mathbf{x}\|_{\mathbf{A}} = \sqrt{\mathbf{x}^\top \mathbf{A} \mathbf{x}}$. For all the functions $f$ appearing in this paper, we simply assume that $f \in \mathcal{C}^2$, which means that $f$ has a well-defined second-order derivative. We use $\nabla f(\mathbf{x})$ and $\nabla^2 f(\mathbf{x})$ to denote the first-order and second-order derivative of $f$. Moreover, we always assume that the objective function $f$ satisfies some basic assumptions in Section 2 and the minimizer of $f$ exists. We use $\mathbf{x}^*$ to denote the minimizer, i.e. $\mathbf{x}^* \triangleq \operatorname{argmin}_{\mathbf{x}} f(\mathbf{x})$ and $f^*$ to denote its minimum value, i.e. $f^* \triangleq \min_{\mathbf{x}} f(\mathbf{x})$.

## 2 PRELIMINARY

In this section, we formally present some definitions and assumptions to constrain the objective function and the optimization problem.

**Assumption 2.1** ($L$-smoothness). *We say a function $f$ is $L$-smooth (or has $L$-Lipschitz continuous gradients), if $\|\nabla f(\mathbf{x}) - \nabla f(\mathbf{y})\| \leq L\|\mathbf{x} - \mathbf{y}\|$ for all $\mathbf{x}, \mathbf{y} \in \mathbb{R}^d$.*

Consequently, for the function $f \in \mathcal{C}^2$, we have the following inequality based on the $L$-smoothness of $f$ (see Nesterov (2003, Chapter 1)): $f(\mathbf{y}) \leq f(\mathbf{x}) + \langle \nabla f(\mathbf{x}), \mathbf{y} - \mathbf{x} \rangle + \frac{L}{2}\|\mathbf{x} - \mathbf{y}\|^2, \quad \forall \mathbf{x}, \mathbf{y} \in \mathbb{R}^d$.

**Assumption 2.2** (Convexity). *We say a function $f$ is convex if $f(\mathbf{y}) \geq f(\mathbf{x}) + \langle \nabla f(\mathbf{x}), \mathbf{y} - \mathbf{x} \rangle + \frac{\mu}{2}\|\mathbf{x} - \mathbf{y}\|^2$ for all $\mathbf{x}, \mathbf{y} \in \mathbb{R}^d$, where $\mu \geq 0$. Moreover, if $\mu > 0$, $f$ is said to be $\mu$-strongly convex.*

**Assumption 2.3** ($H$-Hessian Lipschitz continuity). *We say $f \in \mathcal{C}^2$ has $H$-Hessian Lipschitz continuous Hessian matrices if $\|\nabla^2 f(\mathbf{x}) - \nabla^2 f(\mathbf{y})\| \leq H\|\mathbf{x} - \mathbf{y}\|$ for all $\mathbf{x}, \mathbf{y} \in \mathbb{R}^d$.*

Next we define some frequently-used criteria for an approximate solution. For convex problems, we aim to find an $\epsilon$-approximate solution satisfying the definition below:

**Definition 2.4** ($\epsilon$-approximate solution). *We say $\mathbf{x}$ is an $\epsilon$-approximate solution of $f$ if $f(\mathbf{x}) - f^* \leq \epsilon$.*

For non-convex problems, finding an $\epsilon$-approximate solution in general is NP-hard (Murty & Kabadi, 1985). Instead we consider finding an $\epsilon$-approximate first-order stationary point satisfying the definition below:

**Definition 2.5** ($\epsilon$-stationary point). *We say $\mathbf{x}$ is an $\epsilon$-appriximate first-order stationary point of $f$ if $\|\nabla f(\mathbf{x})\| \leq \epsilon$.*

## 3 COMMON RANDOM RECONSTRUCTION: CORE IDEA

In this section, we present in detail the underlying idea of our Common RandOm REconstruction (CORE) technique behind the algorithm design. We can see such a technique reduces the quantities of data transmitted during communication to a great extent, which significantly reduces the communication complexity. It is of great importance in distributed optimization tasks.

In most distributed machine learning tasks, information is transferred from one machine to another one in vector form, i.e. the gradient of the objective function. Suppose the dimension of the information is $d$. When a machine transmits a $d$-dimensional vector to another machine, the communication cost is $d$. However, in most applications, the dimension $d$ is very large. As a result, it is very expensive to send the whole vector. Inspired by the theory of communication complexity (Andrew, 1979), we propose a **feasible technique which realizes the dimension reduction by randomization**. Specifically, we suppose that all the machines have a common random number generator, which generates a fresh random Gaussian vector $\boldsymbol{\xi} \sim N(0, \mathbf{I}_d)$ at each transmission. We denote the information we want to transmit by $\mathbf{a} \in \mathbb{R}^d$. Instead of sending the $d$-dimension vector $\mathbf{a}$, we send a scalar $\langle \mathbf{a}, \boldsymbol{\xi} \rangle$ which is the inner production of $\mathbf{a}$ and the common random Gaussian vector $\boldsymbol{\xi}$. Then the receiver reconstructs $\mathbf{a}$ by multiplying $\boldsymbol{\xi}$ with the scalar.

---

**Algorithm 1** CORE: Common Random Reconstruction

---

**Require:** An vector $\mathbf{a}$, machines $M_1$ and $M_2$, one-round communication budget $m$, a common random number generator
  **while** $M_1$ want to send $\mathbf{a}$ to $M_2$ **do**
    Generate fresh i.i.d. random Gaussian vectors $\boldsymbol{\xi}_1, \cdots, \boldsymbol{\xi}_m \sim N(0, \mathbf{I}_d)$ with the common random number generator
    $M_1$ sends $\{p_i\}_{i=1}^m$ to $M_2$ with $p_i = \langle \mathbf{a}, \boldsymbol{\xi}_i \rangle$
    $M_2$ reconstructs $\mathbf{a}$ by $\tilde{\mathbf{a}} = \frac{1}{m} \sum_{i=1}^m p_i \cdot \boldsymbol{\xi}_i$
  **end while**

---

To ensure the training accuracy and convergence rate, we can take $m$ fresh random Gaussian vectors for dimension reduction, where $m$ is the one-round communication budget. Specifically, We send $m$ scalars which are the inner products of $\mathbf{a}$ with $m$ random Gaussian vectors, and reconstruct $\tilde{\mathbf{a}}$ by averaging over the reconstructions using all $m$ random Gaussian vectors. We call this compression and reconstruction scheme Common Random Reconstruction (CORE), and describe it in Algorithm 1. In Algorithm 1, the estimation of $\mathbf{a}$ admits:

$$\tilde{\mathbf{a}} = \frac{1}{m} \sum_{i=1}^m \langle \mathbf{a}, \boldsymbol{\xi}_i \rangle \cdot \boldsymbol{\xi}_i. \tag{3}$$

The next important question is whether this technique can guarantee the accuracy of the results. In Lemma 3.1 and Lemma 3.2, we show that $\tilde{\mathbf{a}}$ is an unbiased estimator, and the variance of $\tilde{\mathbf{a}}$ can be bounded under arbitrary matrix norms.

**Lemma 3.1.** $\tilde{\mathbf{a}}$ *is an unbiased estimator of* $\mathbf{a}$:

$$\mathbb{E}_{\boldsymbol{\xi}_1, \cdots \boldsymbol{\xi}_m} \tilde{\mathbf{a}} = \mathbf{a}. \tag{4}$$

**Lemma 3.2.** *The variance of* $\tilde{\mathbf{a}}$ *under norm* $\|\cdot\|_{\mathbf{A}}$, *where* $\mathbf{A}$ *is a given positive semi-definite symmetric matrix, can be bounded by* $\frac{3\mathrm{tr}(\mathbf{A})}{m} \|\mathbf{a}\|^2 - \frac{1}{m} \|\mathbf{a}\|_{\mathbf{A}}^2$:

$$\mathbb{E}_{\boldsymbol{\xi}_1, \cdots, \boldsymbol{\xi}_m} \|\tilde{\mathbf{a}} - \mathbf{a}\|_{\mathbf{A}}^2 \leq \frac{3\mathrm{tr}(\mathbf{A})}{m} \|\mathbf{a}\|^2 - \frac{1}{m} \|\mathbf{a}\|_{\mathbf{A}}^2. \tag{5}$$

**Remark 3.3.** *Lemmas 3.1 and 3.2 bound the first and second moments of* $\tilde{\mathbf{a}}$, *which provide us theoretical guarantee of the convergence accuracy if we replace* $\mathbf{a}$ *by* $\tilde{\mathbf{a}}$ *in certain algorithms. First, it is obvious that* $\tilde{\mathbf{a}}$ *has a **sub-exponential tail** distribution given* $\mathbf{a}$, *so we can provide high probability results using concentration inequalities. Second, the variance of* $\tilde{\mathbf{a}}$ *is **upper bounded** when* $\mathrm{tr}(\mathbf{A})$ *is smaller, ensuring the convergence accuracy of our technique with a lower communication cost.*

In most cases, when eigenvalues decrease rapidly indicating that $\mathrm{tr}(\mathbf{A})$ is not large, our technique demonstrates substantial improvement. Indeed, the CORE technique finds application in a diverse range of distributed optimization tasks across various settings. These include scenarios involving gradient-based algorithms, proximal algorithms, as well as both centralized and decentralized distributed optimization approaches. In this paper, we focus on the gradient-based distributed optimization algorithms on the centralized distributed optimization, by transmitting the reconstruction by our CORE method, $\tilde{\mathbf{a}}$, instead of the full gradient vector $\tilde{\mathbf{a}}$, to reduce the communication cost in each round.

## 4 CORE ON LINEAR MODELS

In this section, we delve into the behavior of CORE on linear models. To provide a clear illustration of the CORE technique, we focus on representative and straightforward cases that encompass the linear model. This model stands as one of the most crucial applications of convex optimization in machine learning. We extend our analysis to more general cases in Section 5 and Apendix D.

---

**Algorithm 2** CORE-GD with per-round communication budget $m$

---

**Require:** $n$ machines, a central machine, a common random number generator, $m \leq \frac{\text{tr}(\mathbf{A})}{L}$, $\mathbf{x}^0$,
    $k = 0$, step-size $h_k = \frac{m}{4\text{tr}(\mathbf{A})}$
  **while** $k < N$ **do**
    Generate fresh i.i.d. $m$ Gaussian vectors $\boldsymbol{\xi}_1, \cdots, \boldsymbol{\xi}_m$ with the common random number generator
    Machine $i$ sends $p_{ij} = \langle \nabla f_i(\mathbf{x}^k), \boldsymbol{\xi}_j \rangle$ to the central machine
    The central machine sends $\sum_{i=1}^{n} p_{ij}$ back to every machine
    Machines reconstruct $\tilde{\nabla}_m f(\mathbf{x}^k)$ by $\tilde{\nabla}_m f(\mathbf{x}^k) = \frac{1}{m} \sum_{i=1}^{n} \sum_{j=1}^{m} p_{ij} \boldsymbol{\xi}_j$
    Machines update $\mathbf{x}^k$ by $\mathbf{x}^{k+1} = \mathbf{x}^k - h_k \tilde{\nabla}_m f(\mathbf{x}^k)$
    $k \leftarrow k + 1$
  **end while**

---

We start with the general components of CORE. Suppose we have $n$ machines. Based on the analysis of our core idea, we use Algorithm 1 to compress and reconstruct the gradient vector as below,

$$\tilde{\nabla}_m f(\mathbf{x}) = \frac{1}{nm} \sum_{i=1}^{n} \sum_{j=1}^{m} \langle \nabla f_i(\mathbf{x}), \boldsymbol{\xi}_j \rangle \cdot \boldsymbol{\xi}_j. \tag{6}$$

Then from Lemma 3.1 and Lemma 3.2, $\tilde{\nabla}_m f(\mathbf{x})$ is an unbiased stochastic estimation of $\nabla f(\mathbf{x})$ with a controlled variance. This implies that if one can design a variety of optimization algorithms using the stochastic oracle $\tilde{\nabla}_m f(\mathbf{x})$, then these algorithms can be efficiently implemented by CORE. In this paper, we introduce two typical algorithms based on GD and AGD.

Now we introduce the CORE-GD algorithm, where at each gradient descent step, the gradient $\nabla f(\mathbf{x})$ is replaced by estimator $\tilde{\nabla} f(\mathbf{x})$ using CORE. The whole algorithm is presented in Algorithm 2, where we let $m$ be the communication budget for a communication round. To show the strength of CORE, we consider the objective function satisfying a mild assumption: $\mathbf{A}$-Hessian domination condition, which is defined as follows:

**Definition 4.1** ($\mathbf{A}$-Hessian domination). *$f$ is said to be $\mathbf{A}$-Hessian dominated if there exists $\mathbf{A}$ such that*

$$\nabla^2 f(\mathbf{x}) \preceq \mathbf{A} \tag{7}$$

*for every $\mathbf{x} \in \mathbb{R}^d$.*

We aim to characterize the complexity in terms of $\text{tr}(\mathbf{A})$. We note that when $f$ is $L$-smooth, a loose bound for $\mathbf{A}$ is $\mathbf{A} \preceq LI$. The fact implies that $\text{tr}(\mathbf{A})$ will reach $dL$ in the worst case, whereas, $\text{tr}(\mathbf{A})$ can be much smaller than $dL$ in most cases. We will show that the linear models are $\mathbf{A}$-Hessian dominated. Moreover, when the data is normalized to a constant level, $\text{tr}(\mathbf{A})$ is much smaller and dimension-free. This result suggests only transmitting $\mathcal{O}(1)$-bits information using CORE without lowering the convergence rate in expectation under suitable conditions. We shall mention that a similar idea of Hessian domination is also considered by Freund et al. (2022) in the Langevin sampling algorithm, who instead proposes a squared Hessian domination condition.

We first consider the $\mu$-strongly convex case. Theorem 4.2 below provides a linear convergence results for Algorithm 2.

**Theorem 4.2.** *Suppose $f$ is $\mu$-strongly convex, $L$-smooth, and $\mathbf{A}$-Hessian dominated. Let $h_k = \frac{m}{4\text{tr}(\mathbf{A})}$. Then, under the hyper-parameter setting in Algorithm 2, $\{\mathbf{x}^k\}_{k \in \mathbb{N}}$ satisfy for all $k \geq 0$*

$$\mathbb{E}f(\mathbf{x}^{k+1}) - f^* \leq \left(1 - \frac{3m\mu}{16\text{tr}(\mathbf{A})}\right)\left(f(\mathbf{x}^k) - f^*\right). \tag{8}$$

**Remark 4.3.** *According to Theorem 4.2, our total communication costs are $\mathcal{O}\left(\frac{\text{tr}(\mathbf{A})}{\mu} \log \frac{1}{\epsilon}\right)$ in expectation. As we have mentioned, high probability results can also be obtained with additional logarithmic factors, which we simply omit here.*

**Remark 4.4.** *We compare CORE-GD with the vanilla CGD algorithm which has total communication costs $\mathcal{O}\left(\frac{dL}{\mu} \log \frac{1}{\epsilon}\right)$. CORE-GD achieves provably lower communication costs since we always have*

$\mathrm{tr}(\mathbf{A}) \le dL$ *when ignoring constants. CORE-GD is also better than DIANA (Mishchenko et al., 2019) whose total communication cost is* $\mathcal{O}(d + \frac{dL}{n\mu})$ *when $d$ is extremely larger than $n$. The communication cost remains unchanged under diffent communication budgets $m$. When $m = \Theta\left(\frac{\mathrm{tr}(\mathbf{A})}{L}\right)$, CORE-GD achieves the same number of communication rounds (convergence rate) as those of CGD when ignoring constants. Bigger communication budget cannot accelerate the convergence rate.*

Next we present realizable conditions for linear models that ensure $\mathrm{tr}(\mathbf{A})$ to be small. We consider the objective admits the so-called ridge-separable form Freund et al. (2022):

$$f(\mathbf{x}) \equiv \frac{1}{N} \sum_{i=1}^{N} \sigma_i(\boldsymbol{\beta}_i^\top \mathbf{x}) + \frac{\alpha}{2} \|\mathbf{x}\|^2. \tag{9}$$

Here, we simply consider the $\ell_2$ norm regularizer. It is possible to generalize our results using proximal algorithms for other regularizers. In (9), $\boldsymbol{\beta}_i$ is associated with the data, and $\sigma_i$ is associated with the loss function. We make the following assumptions:

**Assumption 4.5.** *The functions $\sigma_i \in \mathcal{C}^2$ has bounded second derivatives: $\sigma_i'' \le L_0$ for all $i \in [n]$.*

**Assumption 4.6.** *For all $i \in [N]$, then norm of $\boldsymbol{\beta}_i$ is bounded by $R$: $\|\boldsymbol{\beta}_i\|^2 \le R$.*

Note that Assumption 4.6 can be realized by normalizing the data and Assumption 4.5 only requires that the loss functions have a bounded second derivative. We show that $\mathrm{tr}(\mathbf{A})$ is small:

**Lemma 4.7.** *For the objective function in form of (9), under Assumptions 4.5 and 4.6, then $f$ is $\mathbf{A}$-Hessian dominated and $\mathbf{A}$ satisfies*

$$\mathrm{tr}(\mathbf{A}) \le d\alpha + L_0 R. \tag{10}$$

With Lemma 4.7, we show CORE-GD ensures much low communication costs for linear models under suitable conditions.

**Corollary 4.8.** *For the objective function in form of (9), under Assumptions 4.5 and 4.6, with $\mathrm{tr}(\mathbf{A})$ defined in (10), the total communication costs of CORE-GD are $\mathcal{O}\left(\left(d + \frac{L_0 R}{\alpha}\right) \log \frac{1}{\epsilon}\right)$.*

**Remark 4.9.** *From Corollary 4.8, treated $R$ and $L_0$ as constants, the total communication costs of CORE-GD are $\tilde{\mathcal{O}}(d + \alpha^{-1})$, whereas the vanilla CGD requires $\tilde{\mathcal{O}}(d\alpha^{-1})$ communication costs. Here $\alpha^{-1}$ can be considered as the condition number of the objective since $L$ can be $\Theta(1)$. CORE-GD greatly reduces the communication costs by the factor of $\min(d, \alpha^{-1})$.*

We also consider the acceleration of our algorithm. Specifically, we consider Heavy-ball (Polyak, 1964) acceleration for CORE-GD for quadratic objective functions in Appendix B. From Theorem B.1, the total communication costs to find an $\epsilon$-approximate solution in linear regression model for CORE-AGD are $\tilde{\mathcal{O}}\left(\frac{\sum_{i=1}^{d} \lambda_i^{1/2}}{\mu^{1/2}}\right)$, which is better than $\tilde{\mathcal{O}}(d + \frac{dL^{1/2}}{\mu^{1/2}})$ because $\frac{\sum_{i=1}^{d} \lambda_i^{1/2}}{\mu^{1/2}} \le \frac{d^{1/2} \mathrm{tr}(\mathbf{A})}{\mu^{1/2}}$. When $d$ is large and the trace of Hessian is bounded, this result is better than $\tilde{\mathcal{O}}(d + \frac{dL^{1/2}}{\mu^{1/2}})$. The convergenc rate of CORE-AGD is also better than $\tilde{\mathcal{O}}(\frac{\sum_{i=1}^{d} A_{ii}^{1/2}}{\mu^{1/2}})$ because $\sum_{i=1}^{d} \lambda_i^{1/2} \le \sum_{i=1}^{d} A_{ii}^{1/2}$ when $\mathbf{A}$ is semi-definite. Moreover, when $m = \Theta\left(\frac{\sum_{i=1}^{d} \lambda_i^{1/2}}{L^{1/2}}\right)$, CORE-AGD achieves the same number of communication rounds as those of Centralized AGD with ignoring logarithmic factors.

## 5  CORE-GD FOR NON-CONVEX OPTIMIZATION

In this section, we study CORE-GD on general non-convex problems. To explore the information on Hessian matrices, we further assume that $f$ has $H$-Lipschitz continuous Hessian matrices. We will characterize the complexities of our algorithm in terms of $r_1(f)$, which is often much smaller than $dL$ (see Figure 4 taken from Sagun et al. (2016) and empirical results in related papers, e.g. Sagun et al. (2017); Ghorbani et al. (2019); Brock et al. (2018)). For problems where $r_{1/2}$ is bounded, the results are shown in Appendix D.

Apart from linear models, a broader range of learning models exhibit a restricted $r_1(f)$. We illustrate it with the two-layer neural network model presented below:

**Proposition 5.1.** *Define $f(\mathbf{W}, \mathbf{w}) = \mathbf{w}^\top \sigma(\mathbf{W}\mathbf{x})$, where $\sigma$ is the activation function. When $\|\mathbf{x}\|_1 \leq a_1$, $\|\mathbf{w}\| \leq a_2$ and $\sigma''(x) \leq \alpha$, we have $\mathrm{tr}(\nabla^2 f(\mathbf{W}, \mathbf{w}) \leq \alpha a_1 a_2$.*

Moreover, we notice that for many parameterized models, $r_1(f)$ is limited at least when the parameter is close to its optimal solution. The reason is that under weak regular conditions, the fisher information $\mathcal{I}(\theta) = -\mathbb{E}\left[\frac{\partial^2}{\partial\theta^2}\log f(\mathbf{X};\theta)|\theta\right] = \mathbb{E}\left[\left(\frac{\partial}{\partial\theta}\log f(\mathbf{X};\theta)\right)^2|\theta\right]$. So when $\frac{\partial}{\partial\theta}\log f(\mathbf{X};\theta)$ is bounded, $r_1(f)$ is also bounded. This assurance broadens the scope of applications for our results.

We consider the CORE-Gradient Descent algorithm with some adaptations. The algorithm is shown in Algorithm 4 in Appendix C. Specifically, we take a careful choice of the step size, and give the communication costs under two options. Moreover, we add one more comparison step, for example, $\mathbf{x}^{k+1} \leftarrow \mathrm{argmin}_{\mathbf{x}\in\{\mathbf{x}_k, \tilde{\mathbf{x}}_{k+1}\}} f(\mathbf{x})$. The step requires only one more round of communication with $\mathcal{O}(1)$ communication costs. The theoretical results are presented as follows:

**Theorem 5.2.** *Assume that $f(\mathbf{x})$ is $L$-smooth and has $H$-Lipschitz continuous Hessian matrix. With the assumption of $\mathrm{tr}(\nabla^2 f(\mathbf{x})) \leq r_1$ for any $\mathbf{x} \in \mathbb{R}^d$ and $f(\mathbf{x}^0) - f^* \leq \Delta$. Then, under the hyper-parameter setting in Algorithm 4, the following result in expectation*

$$\mathbb{E}f(\mathbf{x}^k) \leq f(\mathbf{x}^0) - \sum_{i=1}^{k}\mathbb{E}\left[\frac{h_i}{2}\|\nabla f(\mathbf{x}^i)\|^2\right] \tag{11}$$

*holds for option II, and holds with probability $1 - \delta$ for option I.*

**Remark 5.3.** *With Theorem 4, we give the convergence rate and total communication costs of CORE-GD.*

- *For Option I, CORE-GD needs $\mathcal{O}\left(\max\left\{\frac{\Delta r_1(f)}{m\epsilon^2}, \frac{\Delta H^{1/2}d^{3/4}}{m^{3/4}\epsilon^{3/2}}\right\}\right)$ rounds to find an $\epsilon$-stationary point with probability $1 - \delta$. The total communication costs of CORE-GD are*

$$\mathcal{O}\left(\max\left\{\frac{\Delta r_1(f)}{\epsilon^2}, \frac{\Delta H^{1/2}d^{3/4}m^{1/4}}{\epsilon^{3/2}}\right\}\right).$$

- *For Option II, CORE-GD needs $\mathcal{O}\left(\max\left\{\frac{\Delta r_1(f)}{m\epsilon^2}, \frac{\Delta^{5/4}L^{1/4}H^{1/2}d^{3/4}}{m^{3/4}\epsilon^2}\right\}\right)$ rounds to find an $\epsilon$-stationary point in high probability. The total communication costs of CORE-GD are*

$$\mathcal{O}\left(\max\left\{\frac{\Delta r_1(f)}{\epsilon^2}, \frac{\Delta^{5/4}L^{1/4}H^{1/2}d^{3/4}m^{1/4}}{\epsilon^2}\right\}\right).$$

**Remark 5.4.** *Let us compare CORE-GD with Option I with vanilla CGD. The communication costs of CGD to find an $\epsilon$-stationary point is $\tilde{\mathcal{O}}\left(dL\Delta\epsilon^{-2}\right)$. Treated $L$, $H$, $\Delta$ as constants, when the per-round communication budget $m = \Theta\left(\frac{\mathrm{tr}(r_1(f))}{L}\right)$, CORE-GD achieves the same number of communication rounds (convergence rate) as those of CGD, CORE-GD with Option I reduces the communication costs by a factor of $\min(dL/r_1, \epsilon^{-0.5}d^{1/4})$ when ignoring logarithmic factors.*

# 6 CONCLUSION

In this paper, we propose the CORE technique to transmit information in distributed optimization which can dramatically reduce communication costs. We propose our CORE technique based on the common random variables, which provably reduce the quantities of information transmitted, and apply CORE to two distributed tasks. We prove that our CORE-based algorithms achieve lower communication costs. And by choosing the proper communication budget $m$, our algorithms can achieve the same number of communication rounds as those of uncompressed algorithms. In a word, CORE provides new insights and opens the door for designing provably better compression methods in distributed optimization.

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
