. One of the representative achievements is the gradient quantization, for example, 1-bit SGD (Seide et al., 2014) and 1-bit Adam (Tang et al., 2021), which heuristically compresses each component of the gradient into an integer that can be encoded in a few bits. On this basis, TernGrad (Wen et al., 2017), QSGD (Alistarh et al., 2017), ECQ-SGD (Wu et al., 2018), ALQ (Faghri et al., 2020), Natural Compression (Horvóth et al., 2022) and DIANA (Mishchenko et al., 2019) further improved the gradient quantization by adding hyperparameters or combining with the adaptive technique to control the compression ratio. Another main technique is gradient sparsification, which transmits the main dimensions of the gradient instead of the whole. Top-K (Wangni et al., 2018; Shi et al., 2019; Jiang & Agrawal, 2018) was the main idea of gradient sparsification which chose the first $k$ larger dimensions of the gradient to transmit. Gradient Dropping (Aji & Heafield, 2017), DGC (Lin et al., 2017), Atomo (Wang et al., 2018) and IBCD (Mishchenko et al., 2020) obtained better results on this basis. In addition, some new techniques based on other ideas have also been developed and achieved good results. For example, PowerSGD (Vogels et al., 2019) proposed a new low-rank gradient compressor. SignSGD (Bernstein et al., 2018; Safaryan & Richtárik, 2019) proposed a sign-based method with simple compression rules. A biased contractive compressor Beznosikov et al. (2020), a general class of unbiased quantization operators (Horváth et al., 2023) and three-point compressors (3PC) (Richtárik et al., 2022) were also proposed as innovative new achievements. However, the second moments of these estimations are often of order $d$, which implies a restriction of the total communication costs.

**Random sketching.** Sketching (Gribonval et al., 2020; Woodruff et al., 2014; Ikonomovska et al., 2007) is a widely-used technique in machine learning, data mining, and optimization, whose core idea is to reduce the scale by a probabilistic data structure to approximate the data to reduce the computation costs. It is worth noticing that some researchers have started to use the sketching technique to reduce communication costs during the process of training. Specifically, Konečnỳ et al. (2016) proposed FedAvg to reduce the communication costs, which uses a random subset of the value of the full gradient to communicate. They call this method sketched update and integrate quantization (before random sketch) in experiments. Jiang et al. (2018) also proposed a quantization-based sketched gradient compression method, which divides the values of the gradient into four buckets bounded by quantiles and encodes them. However, there is a lack of theoretical guarantees in the convergence of these algorithms, and these methods are still based on random sampling and heuristic quantization encoding. Moreover, Ivkin et al. (2019) proposed SKETCHED-SGD, which uses Count Sketch (Charikar et al., 2004) to compress the gradient. They also presented a theoretical analysis of convergence, but compared with vanilla SGD, it requires an $\tilde{\mathcal{O}}(\frac{1}{T} + \frac{d^2}{k^2 T^2} + \frac{d^3}{k^3 T^3})$ (where $d$ is the dimension of gradient and $k$ is a fixed parameter satisfying $k \leq d$) convergence rate. When $d$ is large, it is much worse than SGD. Rothchild et al. (2020) proposed FetchSGD which combines Count Sketch (Charikar et al., 2004) and Top-$k$ (Lin et al., 2017) for $k$-sparsification. It requires the same convergence rate as SGD for non-convex objective functions, but the communication cost is also dependent on $d$ at least. Hanzely et al. (2018) proved that when adding biased estimates on the basis of random matrix sketching, their algorithm achieves a faster convergence rate and can be accelerated. However, they did not come up with a specific sketching method. Moreover, Lee et al. (2019) and Pilanci et al. (2015) proposed some sketched Hessian-based second-order optimization algorithms. In this work, we mainly focus on gradient-based communication-efficient methods.

**Distributed optimization.** Distributed machine learning and optimization have developed rapidly in recent years. In the early years, the main achievements were based on the existing optimization algorithms, such as SGD with mini-batch (Cotter et al., 2011), DSVRG (Lee et al., 2015), EXTRA (Shi et al., 2015) and MSDA (Scaman et al., 2017b). In recent years, it is worth noticing that some compressed distributed optimization methods have been proposed. Inspired by the results of coordinate gradient descent, Safaryan et al. (2021); Hanzely et al. (2018) proposed a compressed gradient descent framework based on the idea of random projection, achieving an $\tilde{\mathcal{O}}(\frac{\sum_{i=1}^{d} M_{ii}^{1/2}}{\mu^{1/2}})$ communication complexity as Allen-Zhu et al. (2016) for M-Hessian dominated and $\mu$-strongly convex function where $M_{ii}$ is the entry in $i$-th row and $i$-th column of the matrix $\mathbf{M}$, but their achievements are lack of a concrete projection compression method. Moreover, some compressed gradient descent algorithms based on compression techniques mentioned above were also proposed, such as DCGD (Khirirat et al., 2018), DIANA (Mishchenko et al., 2019), MARINA (Gorbunov

et al., 2021), DASHA (Tyurin & Richtárik, 2022) and CANITA (Li & Richtárik, 2021). Li et al. (2020) proposed an accelerated distributed algorithm which achieved $\tilde{\mathcal{O}}(d + \frac{dL^{1/2}}{n\mu^{1/2}})$ considering the setting that the dimension $d$ is extremely larger than $n$, whose lower bound is $\tilde{\mathcal{O}}(d + \frac{d^{1/2}L^{1/2}}{\mu^{1/2}})$. It is worth noticing that in practice $d$ is often extremely large. So there is still a lack of a concrete compression technique and corresponding distributed algorithm that achieves low communication complexity when $d$ is large. And our work fills this gap. In addition, error feedback technique (Stich & Karimireddy, 2019; Karimireddy et al., 2019; Tang et al., 2019; Gruntkowska et al., 2022; Richtárik et al., 2021; Fatkhullin et al., 2021) was also widely used in compressed distributed optimization.

**Federated learning.** Federated Learning is another machine learning setting concentrating on communication costs, where the goal is to train a high-quality centralized model while training data remains distributed over a large number of clients each with unreliable and relatively slow network connections. In federated learning communication bandwidth is a dominating bottleneck and how to design a communication-efficient algorithm has been a concern for researchers. In the early years, some federated learning algorithms have been proposed. These methods are often based on local gradient and heuristic gradient compressors to reduce one-step communication cost, such as FedAvg (Konečnỳ et al., 2016), FetchSGD (Rothchild et al., 2020), SKETCHED-SGD (Ivkin et al., 2019), SCAFFOLD (Karimireddy et al., 2020) and FedLin (Mitra et al., 2021). However, the approximation of local gradient often results in a loss of convergence rate. The total communication costs are worse than, or at best matching that of vanilla gradient descent. Recently, some new communication-efficient methods such as Scaffnew (Mishchenko et al., 2022) and GradSkip (Maranjyan et al., 2022) have been proposed to achieve the same communication rounds as the lower bound of smooth and strongly-convex objective functions $\mathcal{O}(\sqrt{\kappa})$, but the total communication costs are still $\mathcal{O}(d)$.

**Random communication complexity.** In theoretical computer science, communication complexity studies the amount of communication needed to solve a problem when input data is distributed among several parties. Communication complexity was first proposed in Andrew (1979). Andrew (1979) also defined randomized protocol and randomized communication complexity. In a randomized protocol, parties are given a common random string as the input to a deterministic protocol. Random protocols can determine the answer in high probability with much less amount of information transmitted, so randomized communication complexity is much lower than deterministic communication complexity in expectation. Inspired by the advantage of randomized protocols over deterministic ones, we designed a random compression method for distributed optimization which is faster in expectation. Newman (1991) proved that any protocol using a common random string can be simulated by a private random string protocol, with an extra $\mathcal{O}(\log n)$ bits.

## B  ACCELERATION OF CORE-GD ON LINEAR MODELS

In the optimization community, the momentum technique is used to accelerate convergence of Gradient Descent. We also design an accelerated CORE-based algorithm. We name the algorithm as CORE-Accelerated Gradient Descent (CORE-AGD). Here we simply consider the objective function to be quadratic, i.e.

$$f(\mathbf{x}) = \frac{1}{2}\mathbf{x}^\top \mathbf{A}\mathbf{x}. \tag{12}$$

This corresponds to picking $\sigma_i$ as quadratic in linear models. We also note that the quadratic function is already very representative, as it is known that most worst-case functions (lower-bound instances) in the convex optimization are exactly quadratic (see e.g. Nesterov (2003, Chapter 2)). As we have mentioned, our analysis can be directly extended to general convex optimization under an additional Hessian smoothness condition combining with higher-order methods. We present our algorithm in Algorithm 3, which is a heavy-ball (Polyak, 1964) based algorithm by replacing the gradient to be its estimation using CORE.

By a careful analysis of the CORE-AGD algorithm, we have the following theorem:

**Theorem B.1.** *For objective in form of* (12)*, let $\{\lambda_i\}_{i=1}^d$ be the eigenvalues of $\mathbf{A}$ with a decreasing order, and denote $L = \lambda_1$, $\mu = \lambda_d$. Under the hyper-parameter setting in Algorithm 3, we have*

$$\mathbb{E}f(\mathbf{x}^N) \leq 400 \left(1 - \frac{1}{57600}\frac{m\mu^{1/2}}{\sum_{i=1}^d \lambda_i^{1/2}}\right)^N \cdot \frac{L}{\mu} \cdot f(\mathbf{x}^0). \tag{15}$$

---

**Algorithm 3** CORE-AGD

---

**Require:** $n$ machines, a central machine, a common random number generator, $m \leq \frac{\text{tr}(\mathbf{A})}{L}$, $x^0$,
$\quad$ $k = 0, \beta \leftarrow \sqrt{h\mu}, h \leftarrow \frac{m^2}{14400^2 (\sum_i \lambda_i^{1/2})^2}$

$\quad$ **while** $k \leq N$ **do**

$\quad\quad$ Generate fresh i.i.d. $m$ Gaussian vectors $\boldsymbol{\xi}_1, \cdots, \boldsymbol{\xi}_m$ with the common random number generator.

$\quad\quad$ Machine $i$ computes $\mathbf{y}^k = \mathbf{x}^k + (1 - \beta)(\mathbf{x}^k - \mathbf{x}^{k-1})$ and sends $p_{ij} = \langle \nabla f_i(\mathbf{y}^k), \boldsymbol{\xi}_j \rangle$ to the central machine.

$\quad\quad$ The central machine sends $\sum_{i=1}^n p_{ij}$ back to every machine.

$\quad\quad$ Machines reconstruct $\tilde{\nabla}_m f(\mathbf{y}^k)$ by

$$\tilde{\nabla}_m f(\mathbf{x}^k) = \frac{1}{m} \sum_{i=1}^n \sum_{j=1}^m p_{ij} \boldsymbol{\xi}_j \tag{13}$$

$\quad\quad$ Machines update $\mathbf{x}_k$ by $\tilde{\nabla}_m f(\mathbf{y}^k)$ as

$$\mathbf{x}^{k+1} = \mathbf{y}^k - h\tilde{\nabla}(\mathbf{y}^k) \tag{14}$$

$\quad$ **end while**

---

In Theorem B.1, if $f$ is not strongly-convex ($\lambda_d = 0$) or $\lambda_d$ is too small, i.e. ($\lambda < \epsilon$), we can also use the reduction technique (see e.g. Lin et al. (2015)) by adding a regularization term. From Theorem B.1, the total communication costs to find an $\epsilon$-approximate solution for CORE-AGD are $\tilde{\mathcal{O}}\left(\frac{\sum_{i=1}^d \lambda_i^{1/2}}{\mu^{1/2}}\right)$. In contrast, the communication costs of CAGD are $\tilde{\mathcal{O}}\left(\frac{d\lambda_1^{1/2}}{\mu^{1/2}}\right)$. Again, we obtain a provably better communication costs because $\frac{\sum_{i=1}^d \lambda_i^{1/2}}{\mu^{1/2}} \leq \frac{dL^{1/2}}{\mu^{1/2}}$ when ignoring logarithmic factors. And when $m = \Theta\left(\frac{\sum_{i=1}^d \lambda_i^{1/2}}{L^{1/2}}\right)$, CORE-AGD achieves the same number of communication rounds (convergence rate) as those of CAGD when ignoring logarithmic factors. We then specify the objective to satisfy the ridge-separable form (9) with $\sigma_i$ being a quadratic function. We have Corollary B.2, which states that CORE-AGD reduces the communication costs by a $\sqrt{\min(d, \alpha^{-1})}$ factor compared with the "worst-case-optimal" CAGD algorithm.

**Corollary B.2.** *For the objective function in form of* (9) *with $\sigma_i$ being a quadratic function, under Assumptions 4.6 with $R$ treated as a constant, the total communication costs of CORE-AGD are $\tilde{\mathcal{O}}\left(d + \frac{\sqrt{dL_0 R}}{\alpha}\right)$.*

## C CORE-GD FOR NON-CONVEX OPTIMIZATION: ALGORITHM

In this section, we present the CORE-GD algorithm for non-convex optimization problem in Algorithm 4. Specifically, we take a careful choice of the step size, and give the communication costs under two options. Moreover, we add one more comparison step, i.e. $\mathbf{x}^{k+1} \leftarrow \text{argmin}_{\mathbf{x} \in \{\mathbf{x}_k, \tilde{\mathbf{x}}_{k+1}\}} f(\mathbf{x})$. The step requires only one more round of communication with $\mathcal{O}(1)$ communication costs.

## D ACCELERATION OF CORE-GD ON GENERIC CONVEX AND NON-CONVEX PROBLEMS

In this section, we focus on generic optimization problems, where the objective function is $L$-smooth and have an $H$-Lipschitz continuous Hessian matrix. We focus on the settings where $\frac{r_{1/2}(f)}{L^{1/2}}$ is small, and obtain optimal communication costs in terms of $\epsilon$ while reducing the dominating dimension term $d$ to $\frac{r_{1/2}(f)}{L^{1/2}}$.

---

**Algorithm 4** CORE-GD in Non-convex Optimization

---

**Require:** $n$ machines, a central machine, a common random number generator, $m \leq \frac{r_1(f)}{L}$, $\mathbf{x}^0$,
$\quad k = 0$, (For Option I, $m > \log\left(\frac{N}{\delta}\right)$)
$\quad$ Assume that $f(\mathbf{x}^0) - f^* \leq \Delta$
$\quad$ **while** $k < N$ **do**
$\quad\quad$ Generate fresh i.i.d. $m$ Gaussian vectors $\boldsymbol{\xi}_1, \cdots, \boldsymbol{\xi}_m$ with the common random number generator
$\quad\quad$ Machine $i$ sends $p_{ij} = \langle \nabla f_i(\mathbf{x}^k), \boldsymbol{\xi}_j \rangle$ to the central machine
$\quad\quad$ The central machine sends $\sum_{i=1}^n p_{ij}$ back to every machine
$\quad\quad$ Machines reconstruct $\tilde{\nabla}_m f(\mathbf{x}^k)$ by $\tilde{\nabla}_m f(\mathbf{x}^k) = \frac{1}{m}\sum_{i=1}^n\sum_{j=1}^m p_{ij}\boldsymbol{\xi}_j$
$\quad\quad$ Let $p = \frac{1}{m}\sum_{i=1}^m\left(\sum_{j=1}^n p_{ij}\right)$
$\quad\quad$ $h_k = \begin{cases} \min\{\frac{m}{16r_1(f)}, \frac{1}{1600}H^{-1/2}p^{-1/2}d^{-3/4}m^{3/4}\}, & \text{Option I} \\ \min\{\frac{m}{16r_1(f)}, \frac{1}{1600}H^{-1/2}(L\Delta)^{-1/4}d^{-3/4}m^{3/4}\}, & \text{Option II} \end{cases}$
$\quad\quad$ $\tilde{\mathbf{x}}^{k+1} = \mathbf{x}^k - h_k\tilde{\nabla}_m f(\mathbf{x}^k)$
$\quad\quad$ $\mathbf{x}^{k+1} \leftarrow \operatorname{argmin}_{\mathbf{x}\in\{\mathbf{x}_k,\tilde{\mathbf{x}}_{k+1}\}} f(\mathbf{x})$
$\quad\quad$ $k \leftarrow k + 1$
$\quad$ **end while**

---

The key observation is that algorithms using CORE to compress and reconstruct gradient information can be implemented by zeroth-order oracles, instead of first-order oracles. Indeed, one can approximately compute $p = \langle \nabla f(\mathbf{x}), \boldsymbol{\xi} \rangle$ by

$$p \approx \frac{f(\mathbf{x} + \rho \cdot \boldsymbol{\xi}) - f(\mathbf{x})}{\rho}, \tag{16}$$

where $\rho \to 0$. Therefore, we can directly adapt the zeroth-order algorithms from (Yue et al., 2023), where the gradient is also estimated using (16). We first introduce the definition of $D$-bounded distance to the optimal solution and $(\epsilon, \delta)$-SSP as follows. Then we have Theorem D.3 and Theorem D.4.

**Definition D.1** ($D$-bounded distance to the optimal solution)**.** *Assume the minimizer of $f$ exists and $\mathbb{X}^*$ is the of all minimizers. Define $D = \inf_{\mathbf{x}^*\in\mathbb{X}^*}\sup\{\|\mathbf{x} - \mathbf{x}^*\| : f(\mathbf{x}) \leq f(\mathbf{x})\}$.*

**Definition D.2** ($(\epsilon, \delta)$-SSP)**.** *$\mathbf{x}$ is said to be an $(\epsilon, \delta)$-approximated second-order stationary point (SSP) of $f$ if it admits: $\|\nabla f(\mathbf{x})\| \leq \epsilon$ and $\nabla^2 f(\mathbf{x}) \succeq -\delta\mathbf{I}$.*

**Theorem D.3.** *Assume the objective function $f$ is convex and has $L$-continuous gradient and $H$-Lipschitz continuous Hessian matrices. Based on Yue et al. (2023, Algorithm 4), one can find an $\epsilon$-approximated solution in*

$$\tilde{\mathcal{O}}\left(\frac{D \cdot r_{1/2}(f)}{\epsilon^{1/2}} + d \cdot D^{6/7}H^{2/7}\epsilon^{-2/7}\right) \tag{17}$$

*communication costs with high probability.*

**Theorem D.4.** *Assume the objective function $f$ is non-convex and has $L$-continuous gradients and $H$-Lipschitz continuous Hessian matrices. Based on Yue et al. (2023, Algorithm 8), one can find an $(\epsilon, \sqrt{H\epsilon})$-Second-order stationary point of $f$ in*

$$\tilde{\mathcal{O}}\left(r_{1/2(f)}H^{1/4}\Delta\epsilon^{-7/4} + dH^{1/2}\Delta\epsilon^{-3/2}\right) \tag{18}$$

*communication costs with high probability.*

**Remark D.5.** *The communication costs in Theorem D.3 and Theorem D.4 match the state-of-the-art results in terms of $\epsilon$, namely $\tilde{\mathcal{O}}(\epsilon^{-1/2})$ for convex problems (Nesterov, 2003) and $\tilde{\mathcal{O}}(\epsilon^{-7/4})$ for non-convex problems (Jin et al., 2017). In the non-convex case, there still exists an $\epsilon^{-1/28}$ gap between the upper bound and the lower bound (Carmon et al., 2021).*

## E  DECENTRALIZED CORE BASED ALGORITHMS

In this section, we consider the decentralized optimization settings. In centralized settings, we assume that all the machines can send the gradient to the central machine. However, if machines can only

---

**Algorithm 5** Decentralized CORE-GD with per-round communication budget $m$

---

**Require:** $n$ machines, a central machine, a common random number generator, $m \leq \frac{\text{tr}(\mathbf{A})}{L}$, $\mathbf{x}^0$, $k = 0$, step-size $h_k = \frac{m}{4\text{tr}(\mathbf{A})}$.

    **while** $k < N$ **do**

        Generate fresh i.i.d. $m$ Gaussian vectors $\boldsymbol{\xi}_1, \cdots, \boldsymbol{\xi}_m$ with the common random number generator.

        Machine $i$ computes projections $p_{ij} = \langle \nabla f_i(\mathbf{x}^k), \boldsymbol{\xi}_j \rangle$ locally. Define $\mathbf{p}_i = [p_{i1} \quad \cdots \quad p_{im}]^\top$.

        Machines solve an $m$-dimensional subproblem with an decentralized optimization algorithm:

$$\mathbf{p} = \underset{\mathbf{x} \in \mathbb{R}^m}{\arg\min} \frac{1}{n} \sum_{i=1}^{n} \frac{1}{2} \|\mathbf{x} - \mathbf{p}_i\|^2. \tag{19}$$

        Denote $p_j$ to be the $j$th coordinate of $\mathbf{p}$. Machines reconstruct $\tilde{\nabla}_m f(\mathbf{x}^k)$ by

$$\tilde{\nabla}_m f(\mathbf{x}^k) = \frac{n}{m} \sum_{j=1}^{m} p_j \boldsymbol{\xi}_j, \tag{20}$$

        Machines update $\mathbf{x}^k$ by

$$\mathbf{x}^{k+1} = \mathbf{x}^k - h_k \tilde{\nabla}_m f(\mathbf{x}^k). \tag{21}$$

        $k \leftarrow k + 1$.

    **end while**

---

send messages to their neighbours, a message will be transmitted several times before it reaches the central machine. In the worst case, the total communication costs will be multiplied by the diameter of the graph. In the decentralized settings, the communication costs usually depend on the gossip matrix $\mathbf{W}$ of the graph. We propose decentralized CORE-GD in Algorithm 5, and analyze its communication costs.

The optimal solution of subproblem (20) is

$$\mathbf{p} = \frac{1}{n} \sum_{i=1}^{n} \mathbf{p}_i. \tag{22}$$

Therefore, we have

$$p_j = \frac{1}{n} \sum_{i=1}^{n} p_{ij}. \tag{23}$$

By solving supproblem (20), we broadcast $p_j$ to every machine in the graph, and each machine can reconstruct the gradient using $p_j$. The Hessian matrix of the objective function in (20) is $\mathbf{I}_m$, so (20) is simple to optimize. GD will find the optimal solution in one step if (20) can be solved locally. The optimal communication costs of solving (20) to accuracy $\epsilon$ is $\mathcal{O}\left(\frac{1}{\sqrt{\gamma}} \log \frac{1}{\epsilon}\right)$, where $\gamma$ is the eigengap of the gossip matrix $\mathbf{W}$ of the graph (see e.g. Scaman et al. (2017a)). Ignoring logarithmic factors, the total communication costs of decentralized CORE-GD are only $\tilde{\mathcal{O}}\left(\frac{1}{\sqrt{\gamma}}\right)$ times more than the communication costs of centralized CORE-GD in the same setting.

## F    DEFERRED PROOFS IN SECTION 3

*Proof of Lemma 3.1.*

$$\begin{aligned} \mathbb{E}_{\boldsymbol{\xi}_1, \cdot, \boldsymbol{\xi}_m} \tilde{\mathbf{a}} &= \mathbb{E}_{\boldsymbol{\xi}_1, \cdots, \boldsymbol{\xi}_m} \left[ \frac{1}{m} \sum_{i=1}^{m} \langle \mathbf{a}, \boldsymbol{\xi}_i \rangle \cdot \boldsymbol{\xi}_i \right] \\ &= \mathbb{E}_{\boldsymbol{\xi}_1} \boldsymbol{\xi}_1 \boldsymbol{\xi}_1^\top \mathbf{a} = \mathbf{I}\mathbf{a} \\ &= \mathbf{a} \end{aligned} \tag{24}$$

$\square$

*Proof of Lemma 3.2.* For the simplicity of notation, we use $\mathbb{E}_{\boldsymbol{\xi}}$ to denote $\mathbb{E}_{\boldsymbol{\xi}_1,\cdots,\boldsymbol{\xi}_m}$.

$$
\begin{aligned}
\mathbb{E}_{\boldsymbol{\xi}}\|\tilde{\mathbf{a}} - \mathbf{a}\|_{\mathbf{A}}^2 &= \mathbb{E}_{\boldsymbol{\xi}}\left\|\frac{1}{m}\sum_{i=1}^m(\langle\mathbf{a},\boldsymbol{\xi}\rangle\cdot\boldsymbol{\xi} - \mathbf{a})\right\|_{\mathbf{A}}^2 \\
&= \mathbb{E}_{\boldsymbol{\xi}}\left[\frac{1}{m^2}\sum_{i=1}^m\left(\mathbf{a}^\top\boldsymbol{\xi}_i\boldsymbol{\xi}_i^\top\mathbf{A}\boldsymbol{\xi}_i\boldsymbol{\xi}_i^\top\mathbf{a} - \mathbf{a}^\top\mathbf{A}\mathbf{a}\right)\right] \\
&= \frac{1}{m}\mathbb{E}_{\boldsymbol{\xi}_1}\mathbf{a}^\top\boldsymbol{\xi}_1\boldsymbol{\xi}_1^\top\mathbf{A}\boldsymbol{\xi}_1\boldsymbol{\xi}_1^\top\mathbf{a} - \frac{1}{m}\|\mathbf{a}\|_{\mathbf{A}}^2.
\end{aligned}
\tag{25}
$$

Let $\mathbf{A} = \mathbf{U}^\top\mathbf{D}\mathbf{U}$ be the eigenvalue decomposition of $\mathbf{A}$ where $\mathbf{D} = \text{diag}\{b_1,\cdots,b_d\}$ is a diagonal matrix, and $\boldsymbol{\zeta} = \mathbf{U}\boldsymbol{\xi}_1$ be a linear transformation of the random variable $\boldsymbol{\xi}_1$. We have

$$
\begin{aligned}
\mathbb{E}_{\boldsymbol{\xi}_1}\left[\boldsymbol{\xi}_1\boldsymbol{\xi}_1^\top\mathbf{A}\boldsymbol{\xi}_1\boldsymbol{\xi}_1^\top\right] &\overset{a}{=} \mathbb{E}_{\boldsymbol{\zeta}}\left[\mathbf{U}^\top\boldsymbol{\zeta}\boldsymbol{\zeta}^\top\mathbf{D}\boldsymbol{\zeta}\boldsymbol{\zeta}^\top\mathbf{U}\right] \\
&= \mathbf{U}^\top\mathbb{E}_{\boldsymbol{\zeta}}\left[\sum_{i=1}^d b_i\boldsymbol{\zeta}_i^2\cdot\boldsymbol{\zeta}\boldsymbol{\zeta}^\top\right]\mathbf{U} \\
&\overset{b}{=} \mathbf{U}^\top\left(\sum_{i=1}^d b_i\cdot\mathbf{I} + 2\mathbf{D}\right)\mathbf{U} \\
&\overset{c}{=} \text{tr}(\mathbf{A})\cdot\mathbf{I} + 2\mathbf{A} \\
&\preceq 3\text{tr}(\mathbf{A})\cdot\mathbf{I}.
\end{aligned}
\tag{26}
$$

In $\overset{a}{=}$, we use $\boldsymbol{\zeta}\sim N(0,\mathbf{I}_d)$ based on the rotational invariance of the standard Gaussian distribution. In $\overset{b}{=}$, we use the second and forth moment of standard Gaussian variables: $\mathbb{E}\boldsymbol{\zeta}_i^2 = 1$ and $\mathbb{E}\boldsymbol{\zeta}_i^4 = 3$. In $\overset{c}{=}$, we use $\text{tr}(\mathbf{U}^\top\mathbf{D}\mathbf{U}) = \text{tr}(\mathbf{U}^\top\mathbf{U}\mathbf{D}) = \text{tr}(\mathbf{D})$. The last inequality of (26) is due to $\text{tr}(\mathbf{A})\cdot\mathbf{I}\succeq\mathbf{A}$. Combining (26) and (25), we have

$$
\mathbb{E}_{\boldsymbol{\xi}_1,\cdots,\boldsymbol{\xi}_m}\|\tilde{\mathbf{a}} - \mathbf{a}\|_{\mathbf{A}}^2 \leq \frac{3\text{tr}(\mathbf{A})}{m}\|\mathbf{a}\|^2 - \frac{1}{m}\|\mathbf{a}\|_{\mathbf{A}}^2.
\tag{27}
$$

$\square$

## G  DEFERRED PROOFS IN SECTION 4

*Proof of Theorem 4.2.* We write the second-order Taylor expansion of $f(\mathbf{x}^{k+1})$ at $\mathbf{x}^k$:

$$
f(\mathbf{x}^{k+1}) \leq f(\mathbf{x}^k) + \langle\nabla f(\mathbf{x}^k),\mathbf{x}^{k+1} - \mathbf{x}^k\rangle + \frac{1}{2}\langle\mathbf{A}(\mathbf{x}^{k+1} - \mathbf{x}^k),\mathbf{x}^{k+1} - \mathbf{x}^k\rangle.
\tag{28}
$$

Combining the updating process of $\mathbf{x}^{k+1}$ and (28), we have

$$
f(\mathbf{x}^{k+1}) \leq f(\mathbf{x}^k) - h_k\langle\nabla f(\mathbf{x}^k),\tilde{\nabla}_m f(\mathbf{x}^k)\rangle + \frac{h_k^2}{2}\|\tilde{\nabla}_m f(\mathbf{x}^k)\|_{\mathbf{A}}^2.
\tag{29}
$$

Taking expectation with respect to $\tilde{\nabla}_m f(\mathbf{x}^k)$ to both sides of (29), using Lemma 3.1, Lemma 3.2 and Definition 4.1, we have

$$
\begin{aligned}
\mathbb{E}f(\mathbf{x}^{k+1}) &\leq f(\mathbf{x}^k) - h_k\|\nabla f(\mathbf{x}^k)\|^2 + h_k^2\left(\frac{3\text{tr}(\nabla^2 f(\mathbf{x}^k))}{2m}\|\nabla f(\mathbf{x}^k)\|^2 + \|\nabla f(\mathbf{x}^k)\|_{\mathbf{A}}^2\right) \\
&\leq f(\mathbf{x}^k) - \left(h_k - h_k^2\left(\frac{3\text{tr}(\mathbf{A})}{2m} + L\right)\right)\|\nabla f(\mathbf{x}^k)\|^2. \\
&\overset{a}{\leq} f(\mathbf{x}^k) - \left(h_k - h_k^2\cdot\frac{5\text{tr}(\mathbf{A})}{2m}\right)\|\nabla f(\mathbf{x}^k)\|^2,
\end{aligned}
\tag{30}
$$

where in $\overset{a}{\leq}$ we use $m \leq \frac{\text{tr}(\mathbf{A})}{L}$. Then, using $\mu$-strongly convex condition, we have

$$
\begin{aligned}
f^* &\geq \min_{\mathbf{y}}\left\{f(\mathbf{x}^k) + \langle\nabla f(\mathbf{x}^k),\mathbf{y} - \mathbf{x}^k\rangle + \frac{\mu}{2}\|\mathbf{y} - \mathbf{x}^k\|^2\right\} \\
&= f(\mathbf{x}^k) - \frac{1}{2\mu}\|\nabla f(\mathbf{x}^k)\|^2.
\end{aligned}
\tag{31}
$$

Combining (30) and (31), we have

$$
\begin{aligned}
\mathbb{E}f(x^{k+1}) - f^* &\leq f(\mathbf{x}^k) - f^* - 2\mu\left(h_k - \frac{5h_k^2\mathrm{tr}(\mathbf{A})}{2m}\right)\left(f(\mathbf{x}^k) - f^*\right) \\
&\overset{a}{=} \left(1 - \frac{3m\mu}{16\mathrm{tr}(\mathbf{A})}\right)\left(f(\mathbf{x}^k) - f^*\right),
\end{aligned}
\tag{32}
$$

where in $\overset{a}{=}$ we use $h_k = \frac{m}{4\mathrm{tr}(\mathbf{A})}$. Thus, we finish the proof of Theorem 4.2. $\qquad\square$

## H   DEFERRED PROOFS IN SECTION 5

In this section, we prove Theorem 5.2 as below.

**Theorem 5.2.** *Assume that $f(\mathbf{x})$ is $L$-smooth and has $H$-Lipschitz continuous Hessian matrix. With the assumption of $\mathrm{tr}(\nabla^2 f(\mathbf{x})) \leq r_1$ for any $\mathbf{x} \in \mathbb{R}^d$ and $f(\mathbf{x}^0) - f^* \leq \Delta$. Then, under the hyper-parameter setting in Algorithm 4, the following result in expectation*

$$
\mathbb{E}f(\mathbf{x}^k) \leq f(\mathbf{x}^0) - \sum_{i=1}^{k}\mathbb{E}\left[\frac{h_i}{2}\|\nabla f(\mathbf{x}^i)\|^2\right]
\tag{33}
$$

*holds for option II, and holds with probability $1 - \delta$ for option I.*

*Proof of Theorem 5.2.* We write the third-order Taylor expansion of $f(\tilde{\mathbf{x}}^{k+1})$ at $\mathbf{x}^k$:

$$
\begin{aligned}
f(\tilde{\mathbf{x}}^{k+1}) \leq\; & f(\mathbf{x}^k) + \langle\nabla f(\mathbf{x}^k), \tilde{\mathbf{x}}^{k+1} - \mathbf{x}^k\rangle + \frac{1}{2}\langle\nabla^2 f(\mathbf{x}^k)(\tilde{\mathbf{x}}^{k+1} - \mathbf{x}^k), \tilde{\mathbf{x}}^{k+1} - \mathbf{x}^k\rangle \\
& + \frac{H}{6}\|\tilde{\mathbf{x}}^{k+1} - \mathbf{x}^k\|^3.
\end{aligned}
\tag{34}
$$

Combining the updating process of $\tilde{\mathbf{x}}^{k+1}$ with (34), we have

$$
f(\tilde{\mathbf{x}}^{k+1}) \leq f(\mathbf{x}^k) - h_k\langle\nabla f(\mathbf{x}^k), \tilde{\nabla}_m f(\mathbf{x}^k)\rangle + \frac{h_k^2}{2}\|\tilde{\nabla}_m f(\mathbf{x}_k)\|_{\nabla^2 f(\mathbf{x}^k)}^2 + \frac{Hh_k^3}{6}\|\tilde{\nabla}_m f(\mathbf{x}_k)\|^3.
\tag{35}
$$

We denote $\mathbb{E}_k[\cdot] = \mathbb{E}[\cdot|x_k]$. Then taking expectation with respect to $\tilde{\nabla}_m f(\mathbf{x}^k)$ to both sides of (35) and using Lemma 3.1 and Lemma 3.2, we have

$$
\begin{aligned}
\mathbb{E}_k f(\tilde{\mathbf{x}}^{k+1}) &\leq f(\mathbf{x}^k) - h_k\|\nabla f(\mathbf{x}^k)\|^2 + \frac{3h_k^2\mathrm{tr}(\nabla^2 f(\mathbf{x}^k))}{2m}\|\nabla f(\mathbf{x}^k)\|^2 + \frac{Hh_k^3}{6}\mathbb{E}_k\|\tilde{\nabla}_m f(\mathbf{x}_k)\|^3 \\
&\leq f(\mathbf{x}^k) - h_k\|\nabla f(\mathbf{x}^k)\|^2 + \frac{3h_k^2 r_1}{2m}\|\nabla f(\mathbf{x}^k)\|^2 + \frac{Hh_k^3}{6}\mathbb{E}_k\|\tilde{\nabla}_m f(\mathbf{x}^k)\|^3.
\end{aligned}
\tag{36}
$$

Now we give an upper bound of $\mathbb{E}_k\|\tilde{\nabla}_m f(\mathbf{x}^k)\|^3$. We suppose the $m$ random Gaussian vectors are $\boldsymbol{\xi}_i$ for $i \in \{1, \cdots, m\}$. And we denote each $\boldsymbol{\xi}_i$ as

$$
\boldsymbol{\xi}_i = \begin{bmatrix}\xi_{i1} \\ \vdots \\ \xi_{id}\end{bmatrix},
\tag{37}
$$

where $\xi_{ij} \sim N(0,1)$ is independent to each other. Then we have

$$
\begin{aligned}
\mathbb{E}_k \|\tilde{\nabla}_m f(\mathbf{x}^k)\|^3 &\leq \left( \mathbb{E}_k \|\tilde{\nabla}_m f(\mathbf{x}^k)\|^6 \right)^{1/2} \\
&\overset{a}{\leq} \left( 64 \|\nabla f(\mathbf{x}^k)\|^6 + 64 \mathbb{E}_k \left\| \tilde{\nabla}_m f(\mathbf{x}^k) - \nabla f(\mathbf{x}^k) \right\|^6 \right)^{1/2} \\
&\overset{b}{=} 8 \|\nabla f(\mathbf{x}^k)\|^3 \cdot \left( 1 + \mathbb{E} \left( \left( \left( \frac{1}{m} \sum_{i=1}^m (\zeta_{i1}^2 - 1) \right)^2 + \sum_{j=2}^d \left( \frac{1}{m} \sum_{i=1}^m \zeta_{i1}\zeta_{ij} \right)^2 \right)^3 \right) \right)^{1/2} \\
&\overset{c}{\leq} 8 \|\nabla f(\mathbf{x}^k)\|^3 \left( 1 + 20000 \frac{d^3}{m^3} \right)^{1/2} \\
&\leq 1600 \frac{d^{3/2}}{m^{3/2}} \|\nabla f(\mathbf{x}^k)\|^3 .
\end{aligned}
\tag{38}
$$

In $\overset{a}{\leq}$, we use the upper bound of the sixth moment as below.

$$
\begin{aligned}
\mathbb{E} \|X\|^6 &\leq \mathbb{E} \left( \|X - \mathbb{E}X\| + \|\mathbb{E}X\| \right)^6 \\
&\leq \mathbb{E} \left( 2 \max \{ \|X - \mathbb{E}X\|, \|\mathbb{E}X\| \} \right)^6 \\
&\leq 64 \mathbb{E} \|X - \mathbb{E}X\|^6 + 64 \|\mathbb{E}X\|^6 .
\end{aligned}
\tag{39}
$$

In $\overset{b}{=}$, we analyse $\left\| \tilde{\nabla}_m f(\mathbf{x}^k) - \nabla f(\mathbf{x}^k) \right\|^2$ as below. Considering the rotation invariance of the standard Gaussian vectors, we can simplify the computation by rotating the coordinate system. For simplicity, we denote $\nabla f(\mathbf{x}^k) = \mathbf{a}$. We can find an orthogonal matrix $\mathbf{U}$ such that $\mathbf{U}\mathbf{a} = \left[ \|\nabla f(\mathbf{x}^k)\|, 0, \cdots, 0 \right]^\top$. Letting $\hat{\mathbf{a}} = \mathbf{U}\mathbf{a}$ and $\zeta_i = \mathbf{U}\xi_i$, we have $\zeta_i \sim N(0, \mathbf{I}_d)$ and we denote $\zeta_i$ as

$$
\zeta_i = \begin{bmatrix} \zeta_{i1} \\ \vdots \\ \zeta_{id} \end{bmatrix},
\tag{40}
$$

where $\zeta_i j \sim N(0,1)$ is also independent to each other. Then we have

$$
\begin{aligned}
\left\| \tilde{\nabla}_m f(\mathbf{x}^k) - \nabla f(\mathbf{x}^k) \right\|^2 &= \left\| \mathbf{U} \left( \tilde{\nabla}_m f(\mathbf{x}^k) - \nabla f(\mathbf{x}^k) \right) \right\|^2 = \left\| \frac{1}{m} \sum_{i=1}^m \left( \mathbf{a}^\top \xi_i \mathbf{U}\xi_i - \mathbf{U}\mathbf{a} \right) \right\|^2 \\
&= \left\| \frac{1}{m} \sum_{i=1}^m \left( (\mathbf{U}\mathbf{a})^\top (\mathbf{U}\xi_i)\mathbf{U}\xi_i - \mathbf{U}\mathbf{a} \right) \right\|^2 \\
&= \left\| \frac{1}{m} \sum_{i=1}^m \left( \hat{\mathbf{a}}^\top \zeta_i \zeta_i - \hat{\mathbf{a}} \right) \right\|^2 \\
&= \|\nabla f(\mathbf{x}^k)\|^2 \left( \left( \frac{1}{m} \sum_{i=1}^m (\zeta_{i1}^2 - 1) \right)^2 + \sum_{j=2}^d \left( \frac{1}{m} \sum_{i=1}^m \zeta_{i1}\zeta_{ij} \right)^2 \right) .
\end{aligned}
$$

In $\overset{c}{\leq}$, we calculate the high-order moment of standard Gaussian distribution. Especially, we have $\mathbb{E}\zeta_i^{2n} = O(1)$ and $\mathbb{E}\zeta_i^{2n+1} = 0$, where $n \in \{1,2,3,4,5,6\}$ ensuring that

$$
\mathbb{E} \left( \left( \frac{1}{m} \sum_{i=1}^m (\zeta_{i1}^2 - 1) \right)^2 + \sum_{j=2}^d \left( \frac{1}{m} \sum_{i=1}^m \zeta_{i1}\zeta_{ij} \right)^2 \right)^3 = O\left( \frac{d^3}{m^3} \right).
\tag{41}
$$

Combining (38) with (36), we have

$$
\mathbb{E}_k f(\tilde{\mathbf{x}}^{k+1}) \leq f(\mathbf{x}^k) - h_k \|\nabla f(\mathbf{x}^k)\|^2 + \frac{3h_k^2 r_1}{2m} \|\nabla f(\mathbf{x}^k)\|^2 + \frac{800 H h_k^3 d^{3/2}}{3m^{3/2}} \|\nabla f(\mathbf{x}^k)\|^3 .
\tag{42}
$$

For option I, we define the event

$$\mathcal{H}_N = \left(p \geq 2\|\nabla f(\mathbf{x}^k)\|, \quad \forall k \leq N-1\right). \tag{43}$$

Let the event $\tilde{\mathcal{H}}_k = \left(p \geq 2\|\nabla f(\mathbf{x}^k)\|\right)$, with $0 \leq k \leq N-1$. Our choice of $m$ ensures for all $0 \leq k \leq N-1$, $\tilde{\mathcal{H}}_k$ occurs with probability at least $1 - \frac{\delta}{N}$. So, we have

$$\mathbf{P}\left(\mathcal{H}_N\right) = \mathbf{P}\left(\bigcap_{k=0}^{N-1} \tilde{\mathcal{H}}_k\right) \geq 1 - \sum_{k=0}^{N-1} \mathbf{P}\left(\tilde{\mathcal{H}}_k^c\right) \geq 1 - \delta. \tag{44}$$

Therefore, with probability at least $1 - \delta$, $p \geq 2\|\nabla f(\mathbf{x}^k)\|$ holds for all $k = 1, \cdots, N-1$. In the high-probability case, we have

$$\frac{3h_k^2 r_1}{2m}\|\nabla f(\mathbf{x}^k)\|^2 \leq \frac{1}{4}h_k\|\nabla f(\mathbf{x}^k)\|^2, \tag{45}$$

and

$$\frac{800 H h_k^3 d^{3/2}}{3m^{3/2}}\|\nabla f(\mathbf{x}^k)\|^3 \leq \frac{1}{4}h_k\|\nabla f(\mathbf{x}^k)\|^2. \tag{46}$$

For option II, By the choice of $h_k$, we also have

$$\frac{3h_k^2 r_1}{2m}\|\nabla f(\mathbf{x}^k)\|^2 \leq \frac{1}{4}h_k\|\nabla f(\mathbf{x}^k)\|^2, \tag{47}$$

and

$$\frac{800 H h_k^3 d^{3/2}}{3m^{3/2}}\|\nabla f(\mathbf{x}^k)\|^3 \leq \frac{1}{4}h_k\|\nabla f(\mathbf{x}^k)\|^2. \tag{48}$$

Therefore, by summing the (42) over $k$ and taking the full expectation, we have

$$\mathbb{E}f(\mathbf{x}^k) \leq f(\mathbf{x}^0) - \sum_{i=1}^{k}\mathbb{E}\left[\frac{h_i}{2}\|\nabla f(\mathbf{x}^i)\|^2\right], \tag{49}$$

which holds with probability $1 - \delta$ for option I and holds for option II. Now we take a deeper discussion.

- For Option I, In the high-probability case, in $N$ iterations, there are at least $N/2$ rounds of $h_k = \frac{m}{16r_1(f)}$ or $N/2$ rounds of $H^{-1/2}p^{-1/2}d^{-3/4}m^{3/4}$, and in every round $\mathbb{E}f(\mathbf{x}^k)$ decreases by $\mathbb{E}\left[\frac{h_k}{2}\|\nabla f(\mathbf{x}^k)\|^2\right]$. Therefore, CORE-GD needs $\mathcal{O}\left(\max\left\{\frac{\Delta r_1(f)}{m\epsilon^2}, \frac{\Delta H^{1/2}d^{3/4}}{m^{3/4}\epsilon^{3/2}}\right\}\right)$ rounds to find an $\epsilon$-stationary point from $\{\mathbf{x}^k\}_{k=0}^{N-1}$ with probability $1 - \delta$. The total communication costs of CORE-GD are $\mathcal{O}\left(\max\left\{\frac{\Delta r_1(f)}{\epsilon^2}, \frac{\Delta H^{1/2}d^{3/4}m^{1/4}}{\epsilon^{3/2}}\right\}\right)$.

- For Option II, in $N$ iterations, there are at least $N/2$ rounds of $h_k = \frac{m}{16r_1(f)}$ or $N/2$ rounds of $H^{-1/2}(L\Delta)^{-1/4}d^{-3/4}m^{3/4}$, and in every round $\mathbb{E}f(\mathbf{x}^k)$ decreases by $\mathbb{E}\left[\frac{h_k}{2}\|\nabla f(\mathbf{x}^k)\|^2\right]$. Therefore, CORE-GD needs $\mathcal{O}\left(\max\left\{\frac{\Delta r_1(f)}{m\epsilon^2}, \frac{\Delta^{5/4}L^{1/4}H^{1/2}d^{3/4}}{m^{3/4}\epsilon^2}\right\}\right)$ rounds to find an $\epsilon$-stationary point from $\{\mathbf{x}^k\}_{k=0}^{N-1}$ in high probability. The total communication costs of CORE-GD are $\mathcal{O}\left(\max\left\{\frac{\Delta r_1(f)}{\epsilon^2}, \frac{\Delta^{5/4}L^{1/4}H^{1/2}d^{3/4}m^{1/4}}{\epsilon^2}\right\}\right)$.

$\square$

# I  DEFERRED PROOFS IN APPENDIX B

*Proof of Theorem B.1.* Before our proof, we propose a useful Lemma taken from Jin et al. (2017).

**Lemma 5.1.** *Let the $2 \times 2$ matrix $\mathbf{A}$ have following form, for arbitrary $a, b \in \mathbb{R}$,*

$$\mathbf{A} = \begin{bmatrix} a & b \\ 1 & 0 \end{bmatrix}. \tag{50}$$

*Letting $\mu_1$, $\mu_2$ denote the two eigenvalues of $\mathbf{A}$, then, for any $t \in \mathbb{N}$,*

$$
\begin{aligned}
[1 \quad 0]\, \mathbf{A}^t &= \left( \sum_{i=0}^{t} \mu_1^i \mu_2^{t-i} \quad - \mu_1 \mu_2 \sum_{i=0}^{t-1} \mu_1^i \mu_2^{t-i-1} \right), \\
[0 \quad 1]\, \mathbf{A}^t &= [1 \quad 0]\, \mathbf{A}^{t-1}.
\end{aligned}
\tag{51}
$$

Now we start our proof. Let $\mathbf{z}^{k+1} = \begin{bmatrix} \mathbf{x}^{k+1} \\ \mathbf{x}^k \end{bmatrix}$. The iterations of CORE-AGD can be written as

$$\mathbf{z}^{k+1} = \begin{bmatrix} (2 - \beta)(\mathbf{I} - h\mathbf{A}) & -(1 - \beta)(\mathbf{I} - h\mathbf{A}) \\ \mathbf{I} & \mathbf{0} \end{bmatrix} \mathbf{z}^k + h\boldsymbol{\epsilon}^k \triangleq \mathbf{B}\mathbf{z}^k + h\boldsymbol{\epsilon}^k, \tag{52}$$

where $\boldsymbol{\epsilon}^k = \begin{bmatrix} (\mathbf{I} - \frac{1}{m}\sum_{i=1}^{m} \boldsymbol{\xi}_i \boldsymbol{\xi}_i^\top)\mathbf{A}\mathbf{y}_k \\ \mathbf{0} \end{bmatrix}$, representing the error of estimating $\nabla f(\mathbf{x}_k)$ with $\tilde{\nabla}_m f(\mathbf{x}_k)$.

By induction on $k$, we have

$$\mathbf{z}^N = \mathbf{B}^N \mathbf{z}^0 + h \sum_{k=0}^{N-1} \mathbf{B}^{N-k-1} \boldsymbol{\epsilon}^k. \tag{53}$$

Without loss of generality, we assume that $\mathbf{x}^* = \mathbf{0}$. We estimate the distance to the optimal solution by the $\mathbf{A}^2$ norm of $\mathbf{x}^k$. To compute $\|\mathbf{x}^k\|_{\mathbf{A}^2}$, we decompose $\mathbf{x}^k$ into eigen-directions of $\mathbf{A}$, and $\mathbf{B}$ can be decomposed into $2 \times 2$ matrices. For an eigen-direction with eigenvalue $\lambda$, the update of AGD can be written as follows:

$$
\begin{aligned}
\begin{bmatrix} x_{k+1} \\ x_k \end{bmatrix} &= \begin{bmatrix} (2 - \beta)(1 - h\lambda) & -(1 - \beta)(1 - h\lambda) \\ 1 & 0 \end{bmatrix} \begin{bmatrix} x_k \\ x_{k-1} \end{bmatrix} + h \begin{bmatrix} \epsilon \\ 0 \end{bmatrix} \\
&\triangleq \mathbf{B}_\lambda \begin{bmatrix} x_k \\ x_{k-1} \end{bmatrix} + h \begin{bmatrix} \epsilon \\ 0 \end{bmatrix}.
\end{aligned}
\tag{54}
$$

Let $\mu_1$ and $\mu_2$ be the eigenvalues of $\mathbf{B}_\lambda$. Let $\mathbf{C} = \begin{bmatrix} \mathbf{A}^2 & \mathbf{0} \\ \mathbf{0} & \mathbf{A}^2 \end{bmatrix}$. By (53), We have

$$\mathbb{E}\|\mathbf{z}^N\|_{\mathbf{C}}^2 \leq 2\|\mathbf{B}^N \mathbf{z}^0\|_{\mathbf{C}}^2 + 2\mathbb{E} \left\| \sum_{k=0}^{N-1} \mathbf{B}^{N-k-1} \boldsymbol{\epsilon}^k \right\|_{\mathbf{C}}^2. \tag{55}$$

For the $\epsilon^k$ terms, we have

$$
\begin{aligned}
& \mathbb{E}\left\|\sum_{k=0}^{N-1} \mathbf{B}^{N-k-1} \epsilon^k\right\|_{\mathbf{C}}^2 \\
&= \sum_{k=0}^{N-1} \mathbb{E}_{\boldsymbol{\xi}}\left\|\mathbf{B}^{N-k-1} \epsilon^k\right\|_{\mathbf{C}}^2 \\
&= \sum_{k=0}^{N-1} \mathbb{E}_{\boldsymbol{\xi}}\left[\mathbf{y}^{k\top} \mathbf{A}^{\top}(\mathbf{I} - \tfrac{1}{m}\sum_{j=1}^m \boldsymbol{\xi}_j \boldsymbol{\xi}_j^{\top}) \quad \mathbf{0}\right](\mathbf{B}^{N-k-1})^{\top} \mathbf{C} \mathbf{B}^{N-k-1}\left[\begin{array}{c}(\mathbf{I} - \tfrac{1}{m}\sum_{j=1}^m \boldsymbol{\xi}_j \boldsymbol{\xi}_j^{\top})\mathbf{A}\mathbf{y}^k \\ \mathbf{0}\end{array}\right] \\
&\stackrel{\text{Lemma 3.2}}{\leq} 3 \sum_{k=0}^{N-1} \operatorname{tr}\left((\mathbf{B}^{N-k-1})^{\top} \mathbf{C}\mathbf{B}^{N-k-1}\right) \cdot \frac{\|\mathbf{y}^k\|_{\mathbf{A}^2}^2}{m}.
\end{aligned}
\tag{56}
$$

In order to estimate $\operatorname{tr}\left((\mathbf{B}^{N-k-1})^{\top}\mathbf{C}\mathbf{B}^{N-k-1}\right)$, we consider blocks of $\mathbf{B}$ with respect to eigen-directions of $\mathbf{A}$. The contribution of an eigen-direction with eigenvalue $\lambda$ in the trace is

$$
\begin{aligned}
& \operatorname{tr}\left((\mathbf{B}_\lambda^{N-k-1})^{\top} \cdot \begin{bmatrix} \lambda^2 & 0 \\ 0 & \lambda^2 \end{bmatrix} \mathbf{B}_\lambda^{N-k-1}\right) \\
&= \lambda^2\left(\left\|[1 \quad 0]\,\mathbf{B}_\lambda^{N-k-1}\right\|^2 + \left\|[0 \quad 1]\,\mathbf{B}_\lambda^{N-k-1}\right\|^2\right)
\end{aligned}
\tag{57}
$$

By Lemma 5.1, the last line in (57) equals to

$$
\begin{aligned}
& \lambda^2\left\|\left[\sum_{i=0}^{N-k-1} \mu_{\lambda,1}^i \mu_{\lambda,2}^{N-k-1-i} \quad -\mu_{\lambda,1}\mu_{\lambda,2}\sum_{i=0}^{N-k-2}\mu_{\lambda,1}^i\mu_{\lambda,2}^{N-k-2-i}\right]\right\|^2 \\
&+ \lambda^2\left\|\left[\sum_{i=0}^{N-k-2} \mu_{\lambda,1}^i \mu_{\lambda,2}^{N-k-2-i} \quad -\mu_{\lambda,1}\mu_{\lambda,2}\sum_{i=0}^{N-k-3}\mu_{\lambda,1}^i\mu_{\lambda,2}^{N-k-3-i}\right]\right\|^2.
\end{aligned}
\tag{58}
$$

Define $a_\lambda = |\mu_{\lambda,1}| = \sqrt{(1-\beta)(1-h\lambda)}$. By the choice of $\beta$, we have $a_\lambda \leq 1 - \frac{\sqrt{h\mu}}{2}$. We have the following equation:

$$
\lambda^2\left\|\left[\sum_{i=0}^{N-k}\mu_{\lambda,1}^i\mu_{\lambda,2}^{N-k-i} \quad -\mu_{\lambda,1}\mu_{\lambda,2}\sum_{i=0}^{N-k-1}\mu_{\lambda,1}^i\mu_{\lambda,2}^{N-k-1-i}\right]\right\|^2 \leq 4\lambda^2(N-k)^2 a_\lambda^{N-k}. \tag{59}
$$

From the definition of $\mathbf{y}^k$ and Cauchy-Schwartz inequality, we have

$$
\|\mathbf{y}^k\|_{\mathbf{A}^2}^2 \leq 8\|\mathbf{x}^k\|_{\mathbf{A}^2}^2 + 2\|\mathbf{x}^{k-1}\|_{\mathbf{A}^2}^2 \leq 8\|\mathbf{z}^k\|_{\mathbf{C}}^2 + 2\|\mathbf{z}^{k-1}\|_{\mathbf{C}}^2. \tag{60}
$$

Therefore,

$$
\begin{aligned}
& \mathbb{E}\left\|\sum_{k=0}^{N-1} \mathbf{B}^{N-k-1}\epsilon^k\right\|_{\mathbf{C}}^2 \\
&\leq 3\sum_{k=0}^{N-1}\sum_{i=1}^{d} 8\lambda_i^2(N-k)^2 a_{\lambda_i}^{N-k} \cdot \frac{\|\mathbf{y}^k\|_{\mathbf{A}^2}^2}{m} \\
&= 24\sum_{i=1}^{d}\sum_{k=0}^{N-1}\lambda_i^2(N-k)^2 a_{\lambda_i}^{N-k} \cdot \frac{\|\mathbf{y}^k\|_{\mathbf{A}^2}^2}{m}
\end{aligned}
\tag{61}
$$

Then we calculate $\|\mathbf{B}^N \mathbf{z}^0\|_{\mathbf{C}}^2$. As $\mathbf{x}_{-1} = \mathbf{x}_0$, the contribution of an eigen-directions of $\mathbf{A}$ to the norm is

$$
\lambda^2 x_\lambda^2 \left\|\mathbf{B}_\lambda^N \begin{bmatrix} 1 \\ 1 \end{bmatrix}\right\|^2, \tag{62}
$$

where $\lambda$ is the eigenvalue, and $x_\lambda$ is the coefficient of the eigen-decomposition of $\mathbf{x}_0$. By Lemma 5.1, we have

$$
\begin{aligned}
\mathbf{B}_\lambda^N \begin{bmatrix} 1 \\ 1 \end{bmatrix} &= \begin{bmatrix} \sum_{i=0}^N \mu_{\lambda,1}^i \mu_{\lambda,2}^{N-i} - \mu_{\lambda,1}\mu_{\lambda,2} \sum_{i=0}^{N-1} \mu_{\lambda,1}^i \mu_{\lambda,2}^{N-1-i} \\ \sum_{i=0}^{N-1} \mu_{\lambda,1}^i \mu_{\lambda,2}^{N-1-i} - \mu_{\lambda,1}\mu_{\lambda,2} \sum_{i=0}^{N-2} \mu_{\lambda,1}^i \mu_{\lambda,2}^{N-2-i} \end{bmatrix} \\
&= \frac{1}{2} \begin{bmatrix} \mu_{\lambda,1}^N + \mu_{\lambda,2}^N + (2 - \mu_{\lambda,1} - \mu_{\lambda,2}) \sum_{i=0}^N \mu_{\lambda,1}^i \mu_{\lambda,2}^{N-i} \\ \mu_{\lambda,1}^{N-1} + \mu_{\lambda,2}^{N-1} + (2 - \mu_{\lambda,1} - \mu_{\lambda,2}) \sum_{i=0}^{N-1} \mu_{\lambda,1}^i \mu_{\lambda,2}^{N-1-i} \end{bmatrix} \\
&= \frac{1}{2} \begin{bmatrix} \mu_{\lambda,1}^N + \mu_{\lambda,2}^N + (2 - \mu_{\lambda,1} - \mu_{\lambda,2}) \frac{\mu_{\lambda,1}^{N+1} - \mu_{\lambda,2}^{N+1}}{\mu_{\lambda,1} - \mu_{\lambda,2}} \\ \mu_{\lambda,1}^{N-1} + \mu_{\lambda,2}^{N-1} + (2 - \mu_{\lambda,1} - \mu_{\lambda,2}) \frac{\mu_{\lambda,1}^N - \mu_{\lambda,2}^N}{\mu_{\lambda,1} - \mu_{\lambda,2}} \end{bmatrix}
\end{aligned}
\tag{63}
$$

The $\frac{2 - \mu_{\lambda,1} - \mu_{\lambda,2}}{\mu_{\lambda,1} - \mu_{\lambda,2}}$ term in (63) can be bounded as follows:

$$
\begin{aligned}
\frac{2 - \mu_{\lambda,1} - \mu_{\lambda,2}}{\mu_{\lambda,1} - \mu_{\lambda,2}} &= \frac{2 - (2 - \beta)(1 - h\lambda)}{\sqrt{(1 - h\lambda)(h\lambda(2 - \beta)^2 - \beta^2)}} \\
&\le \frac{\beta + h\lambda}{\sqrt{\frac{1}{4} \cdot h\lambda}} \\
&\le 2 + \sqrt{h\lambda} \\
&\le 3.
\end{aligned}
\tag{64}
$$

Therefore,

$$
\begin{aligned}
\left\| \mathbf{B}_\lambda^N \begin{bmatrix} 1 \\ 1 \end{bmatrix} \right\|^2 &\le \left( |\mu_{\lambda,1}^{2N}| + |\mu_{\lambda,2}^{2N}| + 9|\mu_{\lambda,1}^{2N+2}| + 9|\mu_{\lambda,2}^{2N+2}| + |\mu_{\lambda,1}^{2N-2}| + |\mu_{\lambda,2}^{2N-2}| + 9|\mu_{\lambda,1}^{2N}| + 9|\mu_{\lambda,2}^{2N}| \right) \\
&\le 40 \left( 1 - \frac{\sqrt{h\mu}}{2} \right)^{2N-2},
\end{aligned}
\tag{65}
$$

and we have

$$
\| \mathbf{B}^N \mathbf{z}^0 \|_{\mathbf{C}}^2 \le 40 \left( 1 - \frac{\sqrt{h\mu}}{2} \right)^{2N-2} \| \mathbf{z}^0 \|_{\mathbf{C}}^2.
\tag{66}
$$

Finally, we use induction to prove that $\mathbb{E}\|\mathbf{z}^N\|_{\mathbf{C}}^2 < 200(1-b)^N \|\mathbf{z}^0\|_{\mathbf{C}}^2$ where $b = 1 - \frac{\sqrt{h\mu}}{4}$. Suppose that for $k < N$, we have $\mathbb{E}\|\mathbf{z}^k\|_{\mathbf{C}}^2 < 200(1-b)^N \|\mathbf{z}_0\|_{\mathbf{C}}^2$. By (55), we have

$$
\mathbb{E}\|\mathbf{z}^0\|_{\mathbf{C}}^2 \le 80 \left( 1 - \frac{\sqrt{h\mu}}{2} \right)^{2N-2} \|\mathbf{z}^N\|_{\mathbf{C}}^2 + 48h^2 \sum_{i=1}^d \sum_{k=0}^{N-1} \lambda_i^2 (N-k)^2 a_{\lambda_i}^{N-k} \cdot \frac{\|\mathbf{y}^k\|_{\mathbf{A}^2}^2}{m}.
\tag{67}
$$

By the definition of $\mathbf{y}^k$ and the assumption for induction, we have

$$
\mathbb{E}\|\mathbf{y}^k\|_{\mathbf{A}^2}^2 \le 2000(1-b)^{N-1} \|\mathbf{z}^0\|_{\mathbf{C}}^2.
\tag{68}
$$

Using the summation result:

$$
\sum_{k=1}^n k^2 a^k < \frac{1}{(1-a)^3},
\tag{69}
$$

we have

$$
\begin{aligned}
\mathbb{E}\|\mathbf{z}^N\|_{\mathbf{C}}^2 &\le 80 \left( 1 - \frac{\sqrt{h\mu}}{2} \right)^{2N-2} \|\mathbf{z}^0\|_{\mathbf{C}}^2 + 96000(1-b)^{N-1} \sum_{i=1}^d \frac{h^2 \lambda_i^2}{\left(1 - \frac{a_{\lambda_i}}{b}\right)^3} \frac{\|\mathbf{z}^0\|_{\mathbf{C}}^2}{m} \\
&\le 80 \left( 1 - \frac{\sqrt{h\mu}}{2} \right)^{2N-2} \|\mathbf{z}^0\|_{\mathbf{C}}^2 + 384000(1-b)^{N-1} \sum_{i=1}^d \sqrt{h\lambda_i} \frac{\|\mathbf{z}^0\|_{\mathbf{C}}^2}{m}.
\end{aligned}
\tag{70}
$$

By $h = \frac{m^2}{14400^2 (\sum_i \lambda_i^{1/2})^2}$, we have

$$\mathbb{E}\|\mathbf{z}^N\|_\mathbf{C}^2 \leq 80 \left(1 - \frac{\sqrt{h\mu}}{2}\right)^{N-1} \|\mathbf{z}^0\|_\mathbf{C}^2 + 40(1-b)^{N-1}\|\mathbf{z}^0\|_\mathbf{C}^2. \tag{71}$$

Therefore, by $h\mu \leq 14400^{-2}$ and induction, we have $\mathbb{E}\|\mathbf{z}^N\|_\mathbf{C}^2 < 200(1-b)^N \|\mathbf{z}_0\|_\mathbf{C}^2$ holds for positive integers $N$.

Finally, we have

$$\begin{aligned}
\|\mathbf{z}^N\|_\mathbf{C}^2 &= (\mathbf{x}^N)^\top \mathbf{A}^2 \mathbf{x}^N + (\mathbf{x}^{N-1})^\top \mathbf{A}^2 \mathbf{x}^{N-1} \\
&\geq \mu \left((\mathbf{x}^N)^\top \mathbf{A} \mathbf{x}^N + (\mathbf{x}^{N-1})^\top \mathbf{A} \mathbf{x}^{N-1}\right) \\
&= 2\mu \left(f(\mathbf{x}^N) + f(\mathbf{x}^{N-1})\right),
\end{aligned} \tag{72}$$

and

$$\begin{aligned}
\|\mathbf{z}^0\|_\mathbf{C}^2 &= 2(\mathbf{x}^0)^\top \mathbf{A}^2 \mathbf{x}^0 \\
&\leq 2L(\mathbf{x}^0)^\top \mathbf{A} \mathbf{x}^0 \\
&= 4Lf(\mathbf{x}^0).
\end{aligned} \tag{73}$$

Therefore,

$$\begin{aligned}
\mathbb{E}f(\mathbf{x}^N) &\leq \frac{1}{2\mu} \cdot 200(1-b)^N \cdot 4Lf(\mathbf{x}^0) \\
&= 400 \cdot \frac{L}{\mu} \cdot \left(1 - \frac{m\mu}{57600 \sum_i \lambda_i^{1/2}}\right)^N \cdot f(\mathbf{x}^0).
\end{aligned} \tag{74}$$

Thus, we finish our proof of Theorem B.1. $\qquad\square$

## J    DIFFERENIAL PRIVACY

### J.1    INTRODUCTION OF DIFFERENTIAL PRIVACY

In distributed machine learning, privacy has attracted increasing attention. In general, people tend to think about whether the machines will reveal information to attackers. However, in this section we study that when information transmitted (for example, $p_{ij}$ in Algorithm 2) is leaked, attackers still has no access to the actual gradient information. Moreover, the privacy argument proposed by our paper is based on the differential privacy (Dwork, 2006). Usually, there is a trade-off between privacy and accuracy. Since random projection is a differential-private operation, our CORE-GD can naturally satisfy certain differential privacy conditions. Below, we introduce basic definitions and our main result in differential privacy.

First we introduce the definition of adjacent vectors and $(\epsilon, \delta)$-differential privacy as below.

**Definition 5.1.** *For two vectors $\mathbf{x}$ and $\mathbf{y}$, we say $\mathbf{x}$ and $\mathbf{y}$ are adjacent if they satisfy*

$$\|\mathbf{x} - \mathbf{y}\| \leq \Delta_1 \|\mathbf{x}\|. \tag{75}$$

**Definition 5.2.** *Given $\epsilon, \delta \geq 0$, letting the output of an algorithm $M$ with input $\mathbf{x}$ be $M(\mathbf{x})$, the algorithm $M$ satisfies the $(\epsilon, \delta)-$differential privacy property if for an distinguishable set of outputs $S$, and each adjacent variances pairs $\mathbf{x}$ and $\mathbf{y}$, it holds that*

$$\mathcal{P}\left(M(\mathbf{x}) \in S\right) \leq \exp(\epsilon)\mathcal{P}(M(\mathbf{y}) \in S) + \delta. \tag{76}$$

Intuitively, the differential privacy of an algorithm ensures that if two data are adjacent, with a high probability, one cannot distinguish them from the outputs of the algorithm. We notice that in CORE-based algorithm, if two gradient vectors are not far from each other, then after a random projection, the results will be undistinguished with a high probability. So our algorithm naturally has a certain privacy guarantee. Specifically, we have the result below.

**Theorem 5.3.** *Under the assumptions and settings in Corollary 4.8, assume that $\Delta_1 < 0.1$. For any $(\epsilon, \delta)$ satisfying $\epsilon = 20\Delta_1 \ln \frac{1}{\delta}$, Algorithm 2 with the released information $p_{ij}$ satisfies $(\epsilon, \delta)$−differential privacy.*

Theorem 5.3 is based on the observation mentioned above. Surprisingly, Theorem 5.3 does not depend on the choice of $m$. We think this is because the random projection is rotational invariant, so the attacker can only learn about the norm of the gradient and have no idea about its direction.

### J.2 PROOF OF THEOREM 5.3

For the convenience of our proofs, we first present some properties of $(\epsilon, \delta)$-differential privacy.

**Definition 5.4.** *For two adjacent variances pairs* **x** *and* **y***, an algorithm $M$ and outputs o, the privacy loss is defined as*

$$\mathcal{L} = \ln \frac{\mathcal{P}(M(\mathbf{x}) = o)}{\mathcal{P}(M(\mathbf{y}) = o)}. \tag{77}$$

**Lemma 5.5.** *$M$ satisfies $(\epsilon, \delta)$−differential privacy if $\mathcal{P}(\mathcal{L} > \epsilon) \leq \delta$.*

*Proof.* Letting $B = \{o : \mathcal{L} > \epsilon\}$, we have

$$\begin{aligned}
\mathcal{P}(M(\mathbf{x}) \in S) &= \mathcal{P}(M(\mathbf{x}) \in S \cap B) + \mathcal{P}(M(\mathbf{x}) \in S - B) \\
&\stackrel{a}{\leq} \mathcal{P}(M(\mathbf{x}) \in B) + \mathcal{P}(M(\mathbf{x}) \in S - B) \\
&\stackrel{b}{\leq} \mathcal{P}(M(\mathbf{x}) \in B) + \exp(\epsilon)\mathcal{P}(M(\mathbf{y}) \in S - B) \\
&\stackrel{c}{\leq} \mathcal{P}(M(\mathbf{x}) \in B) + \exp(\epsilon)\mathcal{P}(M(\mathbf{y}) \in S) \\
&\stackrel{d}{\leq} \exp(\epsilon)\mathcal{P}(M(\mathbf{y}) \in S) + \delta.
\end{aligned} \tag{78}$$

$\square$

In $\stackrel{a}{\leq}$ and $\stackrel{c}{\leq}$, we use the fact that $\mathcal{P}(X \in A_1) \leq \mathcal{P}(X \in A_2)$ if $A_1 \subseteq A_2$. In $\stackrel{b}{\leq}$, we use Definition 5.4. In $\stackrel{d}{\leq}$, we use $\mathcal{P}(\mathcal{L} > \epsilon) \leq \delta$. And by a similar analysis of Lemma 5.5, we have

**Lemma 5.6.** *$M$ satisfies $(\epsilon, \delta)$−differential privacy if $\mathcal{P}(\mathcal{L} < -\epsilon) \leq \delta$.*

Now we analyze the differential privacy of CORE compression. If we use CORE to compress an vector **a**, we project it to $m$ Gaussian vectors $\boldsymbol{\xi}_1, \cdots, \boldsymbol{\xi}_m$, and send the inner products $p_i = \langle \mathbf{a}, \boldsymbol{\xi}_i \rangle$. We define

$$\boldsymbol{\Xi} = [\boldsymbol{\xi}_1 \cdots \boldsymbol{\xi}_m]^\top, \tag{79}$$

and

$$\mathbf{p} = [p_1 \cdots p_m]^\top \in \mathbb{R}^m. \tag{80}$$

Therefore, we have $\mathbf{p} = \boldsymbol{\Xi}\mathbf{a}$. We define the compression as

$$\begin{aligned}
C : \mathbb{R}^d &\to \mathbb{R}^m \\
\mathbf{a} &\mapsto \mathbf{p}.
\end{aligned} \tag{81}$$

Now we study the distribution of $C(\mathbf{a})$ for further analysis.

**Lemma 5.7.** *$C(\mathbf{a}) \sim N\left(\mathbf{0}, \|\mathbf{a}\|^2 \mathbf{I}_m\right)$.*

*Proof.* By the definition of $C$ and the properties of standard Gaussian distribution, $C(\mathbf{a})$ must follows an mean zero Gaussian distribution. We notice that the covariance of $p_i$ and $p_j$ is

$$\mathbb{E}p_i p_j = \mathbb{E}\mathbf{a}^\top \boldsymbol{\xi}_i \boldsymbol{\xi}_j^\top \mathbf{a} = \begin{cases} 0 & i \neq j, \\ \|\mathbf{a}\|^2 & i = j. \end{cases} \tag{82}$$

Therefore, the variance of $C(\mathbf{a})$ is

$$\mathbb{E}C(\mathbf{a})C(\mathbf{a})^\top = \|\mathbf{a}\|^2 \mathbf{I}_m. \tag{83}$$

$\square$

By Lemma 5.5, 5.6 and 5.7, we can start the proof of Theorem 5.3.

*Proof of Theorem 5.3.* To simplify the representation, let $\sigma_1 = \|\nabla f(\mathbf{x}^k)\|_2$ and $\sigma_2 = \|\nabla f'(\mathbf{x}^k)\|_2$, where $\nabla f(\mathbf{x}^k)$ and $\nabla f'(\mathbf{x}^k)$ are adjacent. By Lemma 5.7, we have $C(\nabla f(\mathbf{x}^k)) \sim N(0, \sigma_1^2 \mathbf{I}_m)$ and $C(\nabla f'(\mathbf{x}^k)) \sim N(0, \sigma_2^2 \mathbf{I}_m)$. Based on Definition 5.4, the privacy loss is

$$
\begin{aligned}
\mathcal{L} &= \ln \left( \frac{\sigma_2^m}{\sigma_1^m} \exp \left( \frac{\|\mathbf{p}\|^2}{2} \left( \frac{1}{\sigma_2^2} - \frac{1}{\sigma_1^2} \right) \right) \right) \\
&= \frac{\|\mathbf{p}\|^2}{2} \left( \frac{1}{\sigma_2^2} - \frac{1}{\sigma_1^2} \right) + m \ln \frac{\sigma_2}{\sigma_1}.
\end{aligned}
\tag{84}
$$

If $\sigma_1 > \sigma_2$ we compute the probability of event $\{\mathcal{L} > \epsilon\}$, which is equivalent to

$$
\|\mathbf{p}\|^2 > \frac{2 \left( \epsilon - m \ln \frac{\sigma_2}{\sigma_1} \right)}{\frac{1}{\sigma_2^2} - \frac{1}{\sigma_1^2}}.
\tag{85}
$$

And if $\sigma_1 < \sigma_2$ we compute the probability of event $\{\mathcal{L} < -\epsilon\}$, which is equivalent to

$$
\|\mathbf{p}\|^2 > \frac{2 \left( \epsilon - m \ln \frac{\sigma_1}{\sigma_2} \right)}{\frac{1}{\sigma_1^2} - \frac{1}{\sigma_2^2}}.
\tag{86}
$$

We define

$$
t = \frac{2\epsilon}{\left| \frac{1}{\sigma_2^2} - \frac{1}{\sigma_1^2} \right|},
\tag{87}
$$

so in both cases, we have $\mathcal{P}(\mathcal{L} > \epsilon) \leq \mathcal{P}(\|\mathbf{p}\|^2 > t)$ or $\mathcal{P}(\mathcal{L} < -\epsilon) \leq \mathcal{P}(\|\mathbf{p}\|^2 > t)$. Noticing that $\|\mathbf{p}\|^2$ is the sum of square of independent identically Gaussian distribution, so we have

$$
\|\mathbf{p}\|^2 \sim \sigma_1^2 \chi_m^2,
\tag{88}
$$

where $\chi_m^2$ is chi-square distribution with the degree of freedom $m$. According to tail bound of chi-square distribution, we have

$$
\mathcal{P}(\|\mathbf{p}\|^2 > t) \overset{a}{\leq} \exp \left( -\frac{t}{20\sigma_1^2} \right) \overset{b}{\leq} \delta.
\tag{89}
$$

In $\overset{a}{\leq}$, we use the tail bound of chi-square distribution. If $X \sim \chi_n^2$, then

$$
\mathcal{P}(X > t \cdot 2n) \leq \exp \left( -\frac{t \cdot n}{10} \right).
\tag{90}
$$

In $\overset{b}{\leq}$, we use the definition of $\epsilon$ that $\epsilon = 20\Delta_1 \ln \frac{1}{\delta}$, We have

$$
\begin{aligned}
t &= 40\Delta_1 \ln \frac{1}{\delta} \cdot \frac{\sigma_1^2 \sigma_2^2}{|\sigma_1^2 - \sigma_2^2|} \\
&= 40\Delta_1 \ln \frac{1}{\delta} \cdot \sigma_1^2 \cdot \frac{1}{|\sigma_1^2/\sigma_2^2 - 1|} \\
&\geq 40\Delta_1 \ln \frac{1}{\delta} \cdot \sigma_1^2 \cdot \frac{1}{2\Delta_1} \\
&= 20 \ln \frac{1}{\delta} \cdot \sigma_1^2.
\end{aligned}
\tag{91}
$$

Therefore, we have proven that

$$
\mathcal{P}(\mathcal{L} > \epsilon) \leq \mathcal{P}(\|\mathbf{p}\|^2 > t) \leq \delta, \qquad \sigma_1 > \sigma_2,
\tag{92}
$$

and

$$
\mathcal{P}(\mathcal{L} < -\epsilon) \leq \mathcal{P}(\|\mathbf{p}\|^2 > t) \leq \delta, \qquad \sigma_1 < \sigma_2.
\tag{93}
$$

Based on Lemma 5.5 and 5.6, we obtain that our algorithm satisfies $(\epsilon, \delta)-$differential privacy. Thus we finish the proof of Theorem 5.3.

$\square$

## K    Experiment Description and Discussions

We conduct experiments to test the CORE method. We train the ridge and logistic regressions on the two datasets: MNIST and covtype. Further, we also train the ResNet18 on the two datasets: CIFAR10 and CIFAR100 to test the effect of our method on neural networks. In this section, we only compare our method with some basic compression technique, for example, Gradient Quantization (Seide et al., 2014; Alistarh et al., 2017; Tang et al., 2021; Wen et al., 2017) and Gradient Sparsity (Aji & Heafield, 2017; Lin et al., 2017), to verify that our algorithm works. We also do not use other compensation techniques such as feedback. In the future, we will add more comparison and improvements to get better experimental results.

**Methods.** We have implemented the following three gradient compression methods to compare their convergence rate and communication complexities.

• Gradient Quantization (Seide et al., 2014; Alistarh et al., 2017; Tang et al., 2021; Wen et al., 2017) and . This method compresses each dimension of the gradient to several bits instead of a 32-bit floating-point number to transmit with some techniques of error feedback to reduce the quantization errors. This method can compress the gradient up to 32 times.

• Gradient Sparsity (Aji & Heafield, 2017; Lin et al., 2017). This method only preserves the dimensions that occupy more than a certain proportion of the norm in the gradient to transmit while accumulating other dimensions to the next step. In this step other dimensions are replaced by 0 to sparse the gradient. This method works better on the models with gradients having dominant components.

• CORE. Our method projects the gradient by common Gaussian random vectors in order to realize dimension reduction, which could compress the gradient by a certain multiple.

**Performance Plot.** We design two kinds of performance plots. One uses the number of "passes" of the dataset as the $x$-axis. Note this also reflects the number of communication round since in our experiments the batch-size for all algorithms are the same. Another uses the number of bits the model transmits as the $x$-axis. Both use the training objective distance to the minimum as the $y$-axis.

### K.1    Linear Model

We use the above three methods on the following two datasets downloaded from the LibSVM website (Chang, 2008):

• The MNIST dataset (784 features). One dataset about $1 * 28 * 28$ pixel handwritten $0 - 9$ pictures.

• The covtype dataset (54 features). One dataset about some features of a piece of land and the types of vegetation that grows on it.

We use distributed gradient descent and accelerated gradient descent to optimize the logistic regression and ridge regression on different datasets. Though we do not give convergence analysis for CORE-AGD on logistic regression, we find it works empirically. Considering experiments on a real distributed system typical set the number of machines up to 16 as Alistarh et al. (2017) but some simulation experiments often set the number of machines much bigger, for example 100 as Freund et al. (2022), we set the number of machines $N = 50$ as a compromise. We take the algorithm without any gradient compression as the baseline and select learning rate from $\{10^{-k} : k \in \mathbb{Z}\}$. In most cases the learning rate is not necessary to be very small, but noticing that Gradient Quantization may cause relatively large error in the early stage of training, we set the learning rate of the algorithm using this method smaller to ensure convergence. We also compare the same algorithm with and without momentum.

To make comparison across datasets, we normalize every vector by its Euclidean norm to ensure the Euclidean norm of each vector is 1.

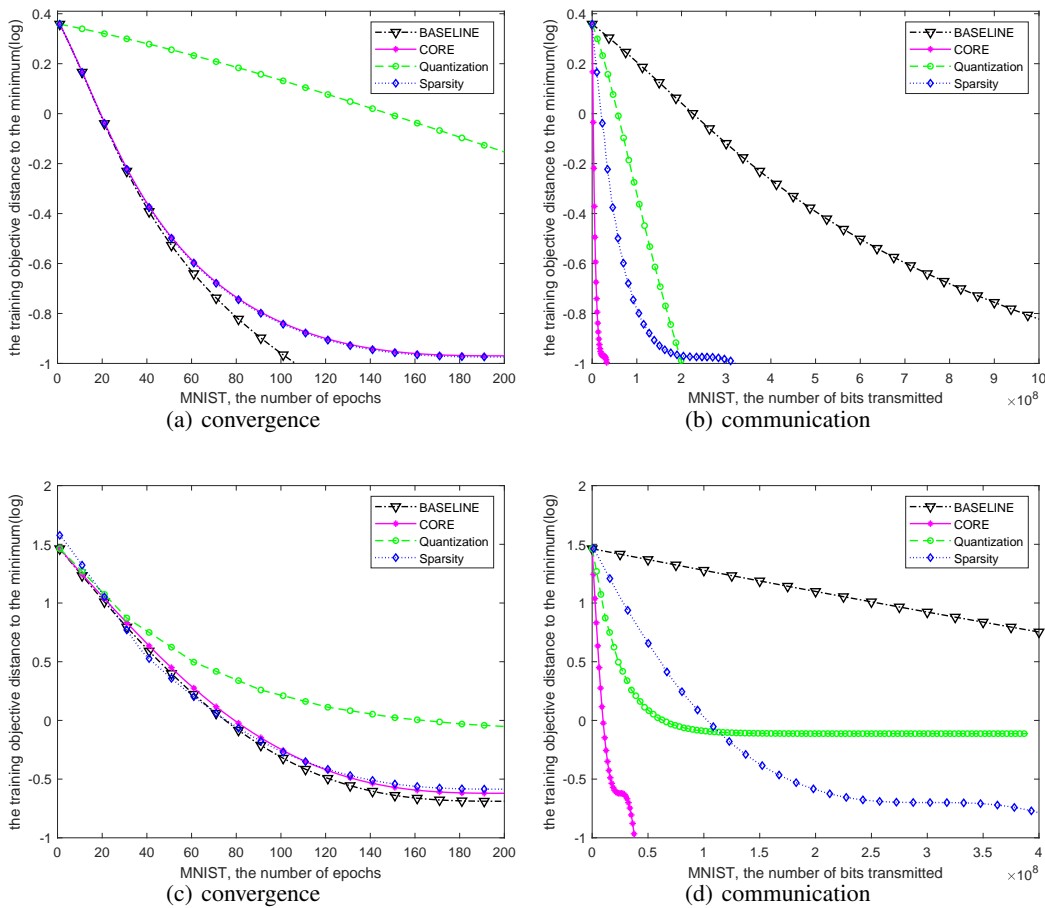

Figure 1: Experiments on MNIST. (a) and (c) plot the function value against the number of epochs respectively, and (b) and (d) plot the function value against communication costs respectively. (a) and (b) plot the result of logistic regression while (c) and (d) plot the result of ridge regression.

The results on linear models are shown in Figure 1 and 2. The results show that our method has lower communication costs while ensuring a nearly same convergence rate (communication round). We notice that the Gradient Quantization has a poor effect with linear models. And compared to the Gradient Sparsity, our method has a significant advantage on communication costs. Another result is that our method works better with momentum.

## K.2  NEURAL NETWORK

We use the above three methods on the following two datasets downloaded from `http://www.cs.toronto.edu/~kriz/cifar.html`:

● the CIFAR10 dataset (50000 samples). One dataset about $3 * 32 * 32$ pixel pictures of 10 kinds of different classes.

● the CIFAR100 dataset (50000 samples). One dataset about $3 * 32 * 32$ pixel pictures of 100 kinds of classes which can be placed into 20 superclasses.

Our goal is to compare our method with the baseline method, Gradient Quantization and Gradient Sparsity on the speed of convergence and communication costs. Moreover, we also compare CORE with some near results such as PowerSGD (Vogels et al., 2019) and DRIVE (Vargaftik et al., 2021). We choose common-used ResNet-18 (He et al., 2016) as the structure of network. We train the model with SGD, the setting of hyperparameters are shown in Table 2.

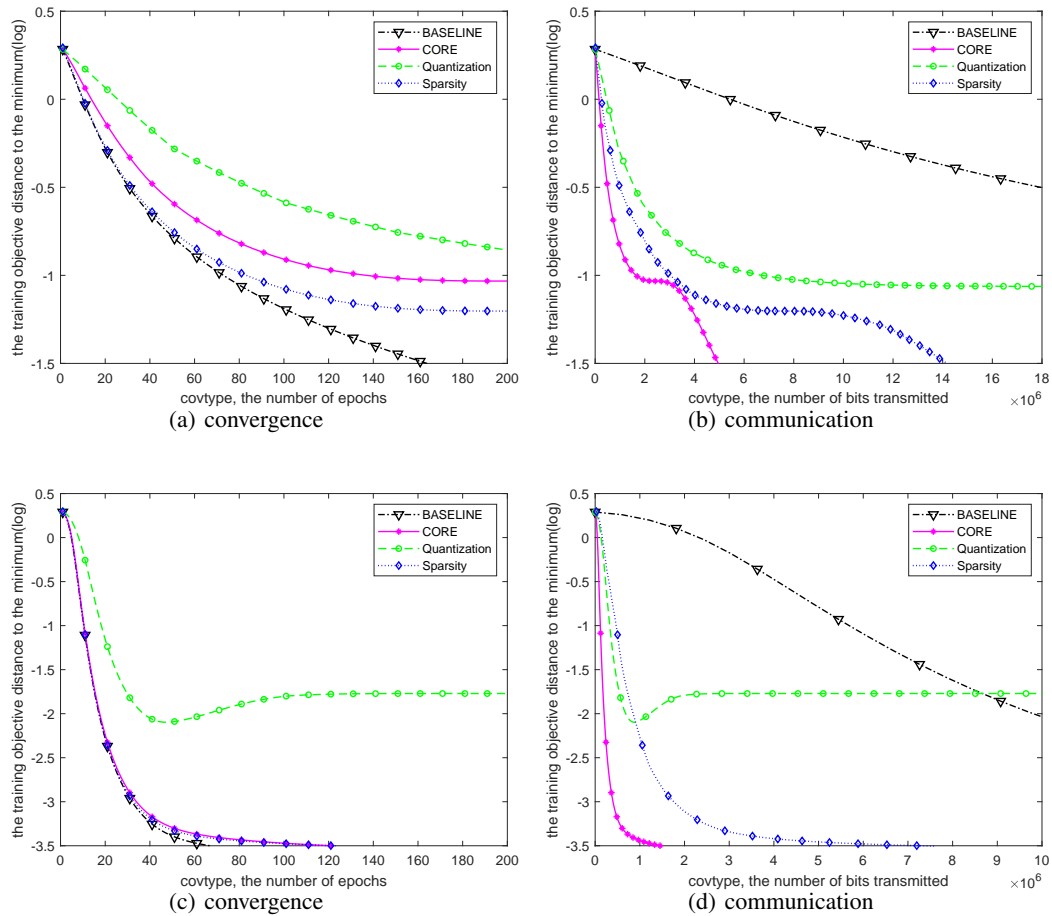

Figure 2: Experiments on covtype with logistic regression. (a) and (c) plot the function value against the number of epochs without and with momentum, respectively, and (b) and (d) plot the function value against communication costs without and with momentum, respectively.

The results on nueral networks are shown in Figure 3. The result shows that our method has a greater convergence rate and communication costs compared to the Gradient Quantization and the Gradient Sparsity. The convergence rate of our method is basically the same as the baseline while the communication costs reduce by hundreds of times. To be more specific, the iteration convergence rate of our CORE method is almost the fastest in the methods participating in the comparison while the number of bits transmitted is much smaller than baseline and almost twice as small as PowerSGD and DRIVE.

## L  ADDITIONS

### L.1  ADDITIONAL FIGURE

We show the eigenvalues of data matrix on MNIST and the eigenvalues of a three-layer neural network on MNIST in Figure 4.

### L.2  MORE MODELS WITH DIMENSION-FREE EFFECTIVE DIMENSION

We will show more learning models for which the effect dimension is dimension-free. As one typical example, we consider the two-layer neural network model under suitable conditions.

**Proposition 5.1.** *Define* $f(\mathbf{W}, \mathbf{w}) = \mathbf{w}^\top \sigma(\mathbf{W}^\top \mathbf{x})$*, where* $\sigma$ *is the activation function. When* $\|\mathbf{x}\|_1 \leq r_1$*,* $\|\mathbf{w}\| \leq r_2$ *and* $\sigma''(x) \leq \alpha$*, we have* $\mathrm{tr}\left(\nabla^2 f(\mathbf{W}, \mathbf{w})\right) \leq \alpha r_1 r_2$*.*

Table 2: Hyperparameter setting of the experiment on networks

| Hyperparameter | CIFAR10 | CIFAR100 |
|---|---|---|
| Batch Size(for all machines) | 1024 | 512 |
| Batch Size(for each machine) | 32 | 16 |
| Machine Numbers | 32 | 32 |
| Optimizer | SGD | SGD |
| Learning Rate | 5e-2 | 5e-2 |
| Min Learning Rate | 3e-6 | 3e-6 |
| Weight Decay | 5e-4 | 5e-4 |
| Epoch | 200 | 200 |
| Learning Rate Scheduler | cosine decay | cosine decay |
| Input Resolution | $32 \times 32$ | $32 \times 32$ |
| Momentum | 0.9 | 0.9 |
| Compression Ratio | 100+ | 80+ |

*Proof.* By direct computation, we have

$$
\begin{aligned}
\frac{\partial f}{\partial \mathbf{w}} &= \sigma(\mathbf{W}^\top \mathbf{x}), \\
\frac{\partial f}{\partial \mathbf{W}} &= \left(\sigma'(\mathbf{W}^\top \mathbf{x}) \odot \mathbf{w}\right) \otimes \mathbf{x}, \\
\frac{\partial^2 f}{\partial \mathbf{w}^2} &= \mathbf{0}, \\
\frac{\partial^2 f}{\partial \mathbf{W}^2} &= \mathrm{Diag}(\sigma''(\mathbf{W}^\top \mathbf{x}) \odot \mathbf{w}) \otimes \mathbf{x} \otimes \mathbf{x}.
\end{aligned}
\tag{94}
$$

Therefore,

$$
\begin{aligned}
\mathrm{tr}\left(\nabla^2 f(\mathbf{W}, \mathbf{w}))\right) &= \|\mathbf{x}\|^2 \cdot \mathrm{tr}\left(\mathrm{Diag}(\sigma''(\mathbf{W}^\top \mathbf{x}) \odot \mathbf{w})\right) \\
&\leq r_1^2 \cdot \langle \sigma''(\mathbf{W}^\top \mathbf{x}), \mathbf{x} \rangle \\
&\leq \alpha r_1 r_2.
\end{aligned}
\tag{95}
$$

$\square$

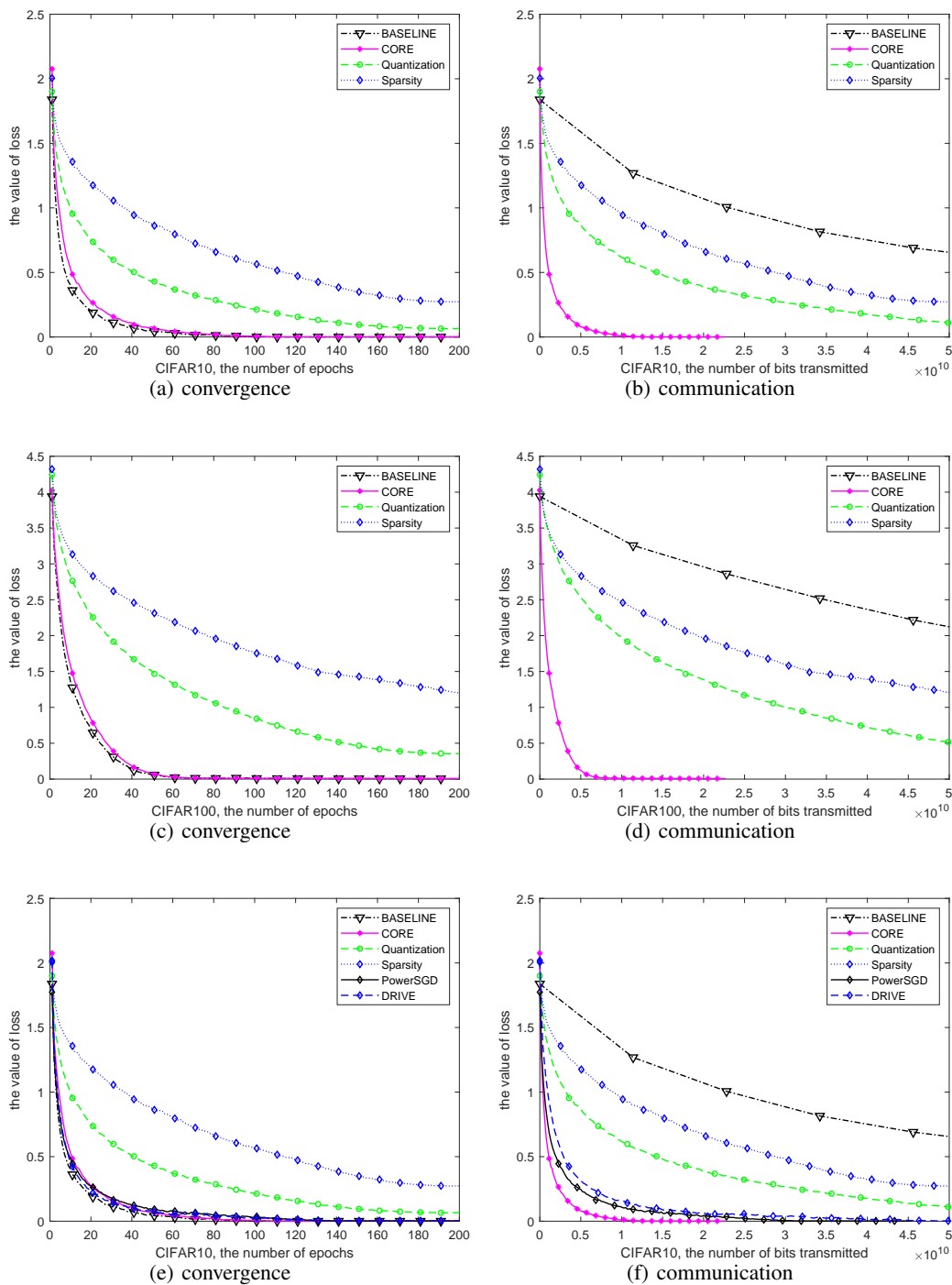

Figure 3: Experiments on the neural network. (a) and (c) plot the function value against the number of epochs on CIFAR10 and CIFAR100, respectively, and (b) and (d) plot the function value against communication costs on CIFAR10 and CIFAR100, respectively. (e) and (f) present more results compared with PowerSGD (Vogels et al., 2019) and DRIVE (Vargaftik et al., 2021).

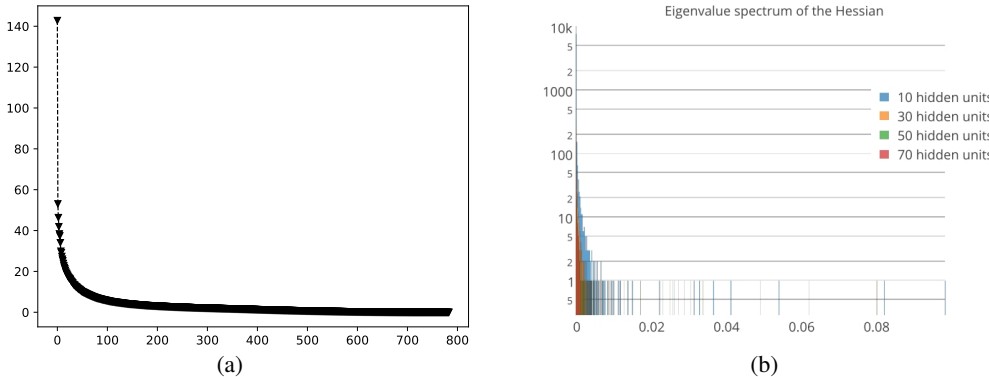

Figure 4: (a) The eigenvalues of the data matrix on MNIST. (b) The eigenvalues of a three-layer neural network on MNIST. (b) is taken directly from Sagun et al. (2016).