# OpenReview forum: "CORE: Common Random Reconstruction for Distributed Optimization with Provable Low Communication Complexity"
_ICLR.cc/2024/Conference — Submitted to ICLR 2024_

### Official Review · Reviewer_pZo4 · 2023-10-25

**Soundness:** 2 fair
**Presentation:** 2 fair
**Contribution:** 1 poor
**Rating:** 3
**Confidence:** 5

**Summary:**

The paper addresses the challenge of minimizing an objective, defined as a finite sum of smooth functions, within a Federated Learning context. The primary concern is the expensive communications encountered in the centralized distributed setting involving multiple nodes or workers.

The authors explore both centralized and decentralized scenarios, focusing on both strongly convex and non-convex loss functions.

To tackle this challenge, a novel gradient compression technique named CORE (Common randOm REconstruction) is introduced. This technique inputs a d-dimensional vector (which can be dense and large) and produces an m-dimensional output, where m is typically much smaller than d. Building on this, the authors propose new methods, namely CORE-GD and CORE-AGD, which leverage the CORE technique for both centralized optimization and decentralized frameworks. They assert that the CORE technique can substantially reduce communication complexity under mild conditions.

The authors support their claims with the following results:
1. Under conditions of $\mu$ strong convexity, $L$-Lipschitz smoothness, and $A$-Hessian domination, they demonstrate the $\mathcal{O}\left(\frac{\operatorname{tr}(A)}{\mu m} \log \frac{1}{\epsilon}\right)$ convergence rate for the CORE-GD method.
2. Within the non-convex setting, given $L$-Lipschitz smoothness, an $H$-Lipschitz continuous Hessian matrix, and an assumption of the uniform boundedness of the Hessian's trace such that $\operatorname{tr}\left(\nabla^2 f(\mathbf{x})\right) \leq r_1$, the authors offer a convergence guarantee to the first order stationary point.
3. Furthermore, they introduce the CORE-Accelerated GD, which is essentially CORE-GD augmented with the heavy-ball trick [1]. For a quadratic objective, the authors suggest that CORE-AGD needs only $\tilde{\mathcal{O}}\left(\frac{\sum_{i=1}^d \lambda_i^{1 / 2}}{\mu^{1 / 2}}\right)$ communications to pinpoint an $\epsilon$-approximate solution.
4. The authors conducted numerical tests, juxtaposing their proposed methods with pre-existing benchmarks.

Reference:
[1] Polyak, B. T. "Some methods of speeding up the convergence of iteration methods." USSR Computational Mathematics and Mathematical Physics, 4(5):1-17, 1964.

**Strengths:**

1) The paper is well-written, with clear main claims and easy readability;

2) A unique compression technique, CORE, is introduced;

3) Through numerical experiments, the authors emphasize the communication efficiency of their proposed methods;

4) The paper offers extensive references to related works on communication compression

**Weaknesses:**

# Suggestions:
- It might be beneficial to incorporate additional citations concerning the Hessian domination condition. Notably, this assumption has been extensively employed in prior coordinate descent methods [3-8], as well as in some recent findings on SGD [2];
- I would recommend that the authors thoroughly review their manuscript, as I encountered a significant number of typos during my reading (see section "Typos" below)
- In the end of this sentence, consider adding some proper citations highliting this fact:
>>With the increase of machine numbers and parameter scale, time spent on communication can not be ignored and even becomes much longer than that on computation"

# Weaknesses
(1) The authors frequently make claims throughout the text without providing supporting evidence. For instance:
1) >> In most cases, when eigenvalues decrease rapidly indicating that tr(A) is not large, our technique demonstrates substantial improvement.

From a mathematical standpoint, even if the Hessian's eigenvalues decrease swiftly, it does not necessarily influence a boundA. As per assumption (7),
$$
\nabla^2 f(\mathbf{x}) \preceq \mathbf{A} \quad \forall \mathbf{x} \in \mathbb{R}^d.
$$
A represents the matrix bound in the worst-case across the function class.

2) >>The fact implies that tr(A) will reach dL in the worst case, whereas, tr(A) can be much smaller than dL in most scenarios. We will show that the linear models are A-Hessian dominated.

The authors have not furnished evidence to support this assertion, neither from a theoretical nor a computational viewpoint.

3) >>Moreover, when the data is normalized to a constant level, tr(A) is much smaller and dimension-free.

For strongly convex situations, the authors present this bound:
$$
\operatorname{tr}(\mathbf{A}) \leq d \alpha+L_0 R .
$$
This does not corroborate the claim that $tr(A)$ is dimension-free.

4) Theorems 4.2 and 5.2 appear to have limited theoretical significance, as it demonstrates a convergence rate in a single-node context. A closer inspection of their proofs reveals its reliance on Lemma 3.2, which sets the variance bounds for the mini-batch estimation $\tilde \nabla_m f(x)$ of $\nabla f(x)$. Notably, the proofs of Theorems 4.2 and 5.2 do not factor in $n$ in any way (nor does the rate). The practical relevance of employing such compression in a single-node scenario is questionable.
Hence, comparison of the rate given by the Theorem 4.2 with DIANA's distributed rate seem misplaced.

5) Regarding the DP results, Theorem 5.3 lacks a comprehensive discussion and appropriate references. Conventionally, in DP papers (refer to [10, 11, 12] for examples), utility guarantees are provided under a chosen noise level, denoted as $\sigma$, which is dependent on the number of iterations. This convention isn't observed in Theorem 5.3.
Furthermore, the initial assumption bounding $\epsilon$ and $\delta$ seems restrictive and not particularly reasonable. If we were to adopt the largest possible value for $\delta$, which is $\delta = \epsilon^{1 / 2}$, this would imply $\epsilon < 0.1$. This corresponds to a relatively high privacy level for the resulting algorithm. Drawing from the classical composition theorem in DP theory, a single CORE-GD step should allocate a privacy budget of at most $0.1 / T$, where $T$ represents the number of iterations at the start of our algorithm. This suggests that even for $T = 100$, there's a need to introduce a substantial amount of noise per iteration, potentially compromising the overall utility.

## Incorrect Information
Table 1 appears to present some inaccuracies:
- Specifically, the DIANA paper by Mishchenko et al. employs Terngrad quantization, not the Top k as indicated.
- A related observation can be made for the ADIANA result. Its convergence guarantees are formulated for the class of unbiased compressors. However, TopK is not a member of this class (refer to [9] for a comprehensive study).

# Typos with corrections and text improvements
## Abstract and introduction
1) >>With distributed machine learning being a prominent technique for large-scale machine learning tasks, communication complexity has become a major bottleneck for speeding up training and scaling up machine numbers

	- Suggestion: Replace "scaling up machine numbers" with "scaling up the number of machines".

2) >>In this paper, we propose a new technique named Common randOm REconstruction (CORE), which can be used to compress the information transmitted between machines in order to reduce communication complexity without other strict conditions.

	- Suggestion: The word "other" seems extraneous. Consider: "...without strict conditions."

3) >>Especially, our technique CORE projects the vector-valued information to a low-dimensional one through common random vectors and reconstructs the information with the same random noises after communication."

The word "Especially" is not used correctly here. Consider replacing it with "Specifically".

4) >> We apply CORE to two distributed tasks, respectively convex optimization on linear models and generic non-convex optimization, and design new distributed algorithms, which achieve provably lower communication complexities."

	- This sentence is a bit convoluted. For clarity, consider breaking it up or restructuring it: "We apply CORE to two distributed tasks: convex optimization on linear models and generic non-convex optimization. We also design new distributed algorithms that achieve provably lower communication complexities."

5) Consider revising "with the convergence rate not worse" to "with a convergence rate that is not worse".
The phrase "preceding the existing results" is unclear. If you mean "improving upon the existing results," it might be clearer to state it that way.

6) >>...guarantees different machines corporately finding...

	- Suggestion: "...guarantees that different machines **cooperatively find**..."

7) >> Typical examples for f i include regression or classification over linear, graphic, as well as (deep) neural network models."

	- Suggestion: "Typical examples for f i include regression or classification over linear, **graph**, as well as (deep) neural network models.

8) >> With the increase of machine numbers and parameter scale, time spent on communication can not be ignored...

	- Correction: "...time spent on communication **cannot** be ignored..."

9) >> However, to the best of our knowledge, although some researches show how to improve existing compression techniques or propose several new ones, few results provide...

	- The word "researches" should be "research."

10) >>...Common randOm REconstruction (CORE) which presents...

	- Insert a comma after "(CORE)"

11) >>Compared with some existing relevant researches, ours has certain advantages.

	- Again, replace "researches" with "research."

12) >> Second, our algorithms provably achieve much lower communication costs compared with the existing algorithms under realizable conditions.

	- For clarity, consider: "Second, under realizable conditions, our algorithms provably achieve much lower communication costs compared to existing algorithms."

13) >> such as quantization (Seide et al., 2014) and sparsity (Aji & Heafield, 2017)

	- replace "sparsity" by "sparsification"

## 1.1 RELATED WORK
1) >>In this section we briefly introduce the related work about our methods...

    - Correction: "In this section, we briefly introduce the related work **on** our methods...

2) >>Federated learning. Federated Learning is another machine learning setting concentrating on communication costs...

    - Consistency: "Federated learning. **Federated learning** is another machine learning setting concentrating on communication costs...

3) >>Random communication complexity. In theoretical computer science, communication complexity studies the amount of communication needed to solve a problem when input data is distributed among several parties."

    - Clarity: "Random communication complexity. **Within** theoretical computer science, communication complexity **refers to the study of** the amount of communication needed to solve a problem when input data is distributed among several parties."

4) >>Random protocols can determine the answer in high probability with much less amount of information transmitted, so randomized communication complexity is much lower than deterministic communication complexity in expectation."

    - Correction: "Random protocols can determine the answer **with high probability** and transmit **a significantly smaller amount** of information, so the randomized communication complexity is expected to be much lower than deterministic communication complexity."

## 1.2 CONTRIBUTIONS
1) >>..To the best of our knowledge, CORE is the first concrete and feasible compression method that achieves a limited bounded variance of the estimate and provably reduce communication complexity...

    - Correction: "... and provably **reduces** communication complexity..."

2) >>Some recent empirical studies (Sagun et al., 2016) carefully compute the eigenvalue of Hessian curves..."

     - Correction: "... carefully **computed** the eigenvalue..."

3) >>This is inspired by the recent work of zeroth-order optimization (Yue et al., 2023), Langevin sampling (Freund et al., 2022), and distributed optimization (Hanzely et al., 2018).

     - Suggestion (for clarity): "This is inspired by the recent work **on** zeroth-order optimization..."

4) >>In summary, the contribution of the paper is listed below:

      - Suggestion (for style): "In summary, the **contributions** of the paper **are** listed below:"

9) >>... achieve the lower bound of iteration rounds the state-of-the-art total communication costs under the realizable condition.

      - Suggestion (for clarity): "... achieve the lower bound of iteration rounds **and** the state-of-the-art total communication costs under the realizable condition."

10) ... where γ is the eigengap of the gossip matrix for the network topology."
      - Suggestion (for clarity): "... where γ is the eigengap of the gossip matrix **associated with** the network topology.

These are just a few samples. I haven't checked the rest of the text for grammatical mistakes in detail.



# References:

[2] Safaryan, et al. “Smoothness matrices beat smoothness constants: Better communication compression techniques for distributed optimization”, In Advances in Neural Information Processing Systems, 2021

[3] Qu, et al. “Coordinate descent with arbitrary sampling I: algorithms and complexity.” Optimization Methods and Software, 2016.

[4] Qu, et al. “Coordinate descent with arbitrary sampling II: expected separable overapproximation.” Optimization Methods and Software, 2016

[5] Hanzely, et al. “Accelerated coordinate descent with arbitrary sampling and best rates for minibatches.” In Proceedings of Machine Learning Research, 2019

[6] Qu, et al. “SDNA: Stochastic Dual Newton Ascent for Empirical Risk Minimization”, In International Conference on Machine Learning, 2015.

[7] Hanzely, et al. “SEGA: Variance reduction via gradient sketching”, In Advances in Neural Information Processing Systems, 2018

[8] Hanzely, et al. “Variance Reduced Coordinate Descent with Acceleration: New Method With a Surprising Application to Finite-Sum Problems.” In International Conference on Machine Learning, 2020

[9] Beznosikov, et al. “On biased compression for distributed learning. arXiv:2002.12410, 2020.”

[10] Lowy, et al. "Private non-convex federated learning without a trusted server". arXiv preprint arXiv:2203.06735, 2022.

[11] Ding, et al. "Diéerentially private and communication eìcient collaborative learning." In Proceedings of the AAAI Conference on Artificial Intelligence, Virtual Conference, 2021.

[12] Zhang, et al. "Private and communication-eìcient edge learning: a sparse diéerential Gaussian-masking distributed SGD approach." In Proceedings of the Twenty-First International Symposium on Theory, Algorithmic Foundations, and Protocol Design for Mobile Networks and Mobile Computing, 2020.

**Questions:**

1) >>To the best of our knowledge, CORE is the first concrete and feasible compression method that achieves a limited bounded variance of the estimate

Could the authors clarify what they mean by "limited bounded variance"?
Additionally, how is this assertion supported theoretically?

2) >>Compared with the results in Hanzely et al. (2018) , our works present a concrete compression technique.

Could the authors clarify what they mean by a "concrete method"?
In my perspective, the Accelerated SEGA algorithm is well-defined, as presented by Hansely et al. 2018.

3) Could the authors elaborate on the transitions in equation (26)? The derivation leading to equation (b) is not clear.

**Details Of Ethics Concerns:**

This work is primarily theoretical in nature.

---

### Official Review · Reviewer_4Rbu · 2023-10-27

**Soundness:** 3 good
**Presentation:** 3 good
**Contribution:** 2 fair
**Rating:** 5
**Confidence:** 4

**Summary:**

This paper introduces CORE, a communication compression technique for distributed optimization. CORE operates like a sketching-type scheme, wherein local gradients undergo encoding using shared random vectors (across both the server and machines) prior to transmission to the server. Subsequently, the server performs decoding of the aggregated message using the same set of shared random vectors. The performance of CORE is demonstrated in its effectiveness in minimizing strongly convex functions, linear models, and select non-convex problems. Particularly, the benefit of CORE is evident when  $\text{tr}(\nabla^2 f)$ is small compared to $dL$.

**Strengths:**

The paper is well-written and the proofs are easy to follow.

The proposed method is easy to implement.

The benefit of the proposed method is evident for the skewed Hessians which is common in practice, suggesting its promising utility.

**Weaknesses:**

1. The obtained total communication cost $\tilde{O}(r_{1/2}(f)/\mu^{1/2})$ is independent of the number of machines $n$, meaning that CORE does not reap advantages from multiple machines collaborating on communication compression. This characteristic likely stems from the shared randomness utilized in CORE's compression process, in contrast to DIANA and ADIANA, which employ independent compressors across machines.  It's worth exploring whether CORE could incorporate independent randomness to harness the benefits of larger values of $n$.


2. The paper does not investigate the impact of gradient stochasticity (i.e., when only stochastic gradients are available on local machines). While it is presumable that CORE would exhibit convergence due to its unbiased estimates, its effect on the asymptotically dominant rate under stochastic gradients remains unclear. In scenarios involving non-convex functions with variances of gradients bounded by $\sigma^2$, does CORE maintain the optimal asymptotic rate of $O(L\sigma/\sqrt{nT})$ (see, e.g., [R1, Theorem 3]) that is independent of compression? Could CORE potentially enhance this rate by improving $L$ to some $r_{\alpha}$?


3. The paper only establishes acceleration for quadratic functions. Is there an inherent reason that acceleration with CORE can be provably guaranteed only for quadratic functions?


4. The authors assert that CORE outperforms ADIANA in the paper. However, the authentic iteration complexity as presented in Li et al. should be $\tilde{O}(\omega+(1+\omega/\sqrt{n}+\omega^{3/4}/n^{1/4})L^{1/2}/\mu^{1/2})$, with the term $\omega^{3/4}/n^{1/4}$ being dominated by $\omega/\sqrt{n}$ when $\omega\geq n$. This rate is further improved in a recent work [R2, Theorem 3], which removes the gap, resulting in $\tilde{O}(\omega+(1+\omega/\sqrt{n})L^{1/2}/\mu^{1/2})$, which is considered minimax optimal. Thus, only $\omega \geq \sqrt{n}$ is needed for ADIANA to attain $\tilde{O}(\omega+\omega/\sqrt{n}L^{1/2}/\mu^{1/2})$, instead of $\omega \geq n$.  To ensure rigor, the authors should also cite R2.
Neverthelesss, given the iteration complexity $\tilde{O}(\omega+(1+\omega/\sqrt{n})L^{1/2}/\mu^{1/2})$, ADIANA implies a total communication cost of $\tilde{O}(d+(K+d/\sqrt{n})L^{1/2}/\mu^{1/2})$ by using random-k operators. Since $r_{1/2}(f) \geq d \mu^{1/2}$, CORE is superior only when $\max\\{K, d/\sqrt{n}\\}L^{1/2}\geq r_{1/2}(f)$. The authors should be more accurate whenever claiming CORE outperforms ADIANA.

R1. Momentum Provably Improves Error Feedback, Fatkhullin, Ilyas and Tyurin, Alexander and Richtarik, Peter, arXiv preprint, 2023.

R2. Unbiased Compression Saves Communication in Distributed Optimization: When and How Much, He, Yutong and Huang, Xinmeng and Yuan, Kun, arXiv preprint, 2023

**Questions:**

Intuitively, I feel CORE provides a preconditioning effect by smoothing highly skewed gradients (i.e., $dL\geq \text{tr}(\nabla^2 f)$) to relatively flat ones governed by the trace $\text{tr}(\nabla^2 f)$. Can authors also compare CORE with the communication cost of the non-compressed gradient descent under the optimal preconditioning, whose rate, I believe, also relates to Hessians?


In summary, I believe that CORE demonstrates significant practical value in deterministic optimization when Hessians are highly skewed. I am open to raise my evaluation if my concerns are adequately addressed.

---

### Official Review · Reviewer_2Q9p · 2023-10-28

**Soundness:** 3 good
**Presentation:** 3 good
**Contribution:** 2 fair
**Rating:** 3
**Confidence:** 3

**Summary:**

The authors propose a compression consisting of projections of the gradient vector on random Gaussian codebooks, where the specific codebook realization needs to be shared between encoder and decoder. Reducing the communication load is relevant in distributed optimization, where gradients need to be exchanged several times between the corresponding nodes. They show that their proposed scheme provides a smaller communication load orderwise than existing schemes if the eigenvalues of the Hessian of the loss function drop fast. They apply their proposed method to two scenarios, to convex optimization for linear models, and generic non-convex optimization.

**Strengths:**

I like the comprehensive summary of all communication-efficient methods with their order results in Table 1 and the comparison with their results. Their proposed CORE scheme provides a larger reduction of the communication complexity for linear models when the eigenvalues of the Hessian of the loss function drop fast.

**Weaknesses:**

Unfortunately, the paper has several weaknesses:
- Essentially, the scheme consists of projecting the gradient vector onto m Gaussian vectors, which has been exploited in form of compressed sensing for a long time with the additional constraint of sparsity. So, the proposed technique is in general not new, which should be highlighted properly in the paper.
- It is not clear to me how the overall communication cost is defined and computed. Shouldn't it depend on the number of Gaussian realizations m in (3), because this is what is exchanged between the clients? Perhaps there is a normalization factor I have been missing.
- The linear convex case for which the authors show some improvements, is not that practically relevant as the authors' claim, as most better performing models are non-linear.
- How are Assumptions 4.5 and 4.6 justified in practice?

The main weakness of the paper is that it only provides order results, which are not practically relevant, as constants can have a big impact. Therefore, it is difficult to compare most of the results in Table 1 in practice. Second, the authors claim that they "provide a concrete and feasible compression technique", but any experimental results and practical comparisons are completely absent which could have helped to undermine the authors' claim. Also, I don't believe that their scheme is practical as the performance highly depends on the specific realizations of the Gaussian vectors they use. To show practicality, the authors would need to show that any choice of random vectors is "good", i.e., it is highly concentrated around the mean, and that the number of "bad" vectors is asymptotically vanishing. Another practical disadvantage is that these vectors need to be shared between all parties. Third, how does classical quantization perform in the context of the scenario at hand? There are multiple papers dealing with distributed learning and quantized gradients, which is the only way to approach this problem if you have resource constraint communication channels.

**Questions:**

Please see the questions I posed in "Weaknesses" above.

---

### Meta-Review · Area_Chair_HfeJ · 2023-12-09

**Metareview:**

The paper has all negative reviews, and there has not been any rebuttal from the authors.

**Justification For Why Not Higher Score:**

The authors have made no attempts to respond to reviews, which are all negative

**Justification For Why Not Lower Score:**

NA

---

### Decision · Program_Chairs · 2024-01-16

Reject